# tRNA lysidinylation is essential for the minimal translation system in the *Plasmodium falciparum* apicoplast

Rubayet Elahi [ID] [1,2][✉] & Sean T Prigge [ID] [1,2][✉]

## Abstract

For decades, researchers have sought to define minimal translation systems to uncover fundamental principles of life and advance biotechnology. tRNAs, essential components of this machinery, decode mRNA codons into amino acids. The apicoplast of malaria parasites contains 25 tRNA isotypes in its organellar genome—the lowest number found in known translation systems. Efficient translation in such minimal systems depends heavily on post-transcriptional tRNA modifications. One such modification, lysidine at the wobble position (C34) of tRNA$_{CAU}$, distinguishes between methionine (AUG) and isoleucine (AUA) codons. tRNA isoleucine lysidine synthetase (TilS) produces lysidine, which is nearly ubiquitous in bacteria and essential for cellular viability. Here, we report a TilS ortholog (*Pf*TilS) targeted to the apicoplast of *Plasmodium falciparum*. We demonstrate that *Pf*TilS activity is essential for parasite survival and apicoplast function, likely due to its role in protein translation. This study is the first to characterize TilS in an endosymbiotic organelle, contributing to research on eukaryotic organelles and minimal translational systems. Moreover, the absence of lysidine in humans highlights a potential target for antimalarial strategies.

**Keywords** Apicoplast; Lysidine; Plasmodium; Protein Translation; tRNA Modification
**Subject Categories** Microbiology, Virology & Host Pathogen Interaction; RNA Biology; Translation & Protein Quality

## Introduction

Elucidating the minimal genome and its translational machinery is fundamental to understanding life and advancing biotechnology. Central to cellular maintenance is the translational machinery, comprising RNA molecules (transfer RNA [tRNA], messenger RNA [mRNA], ribosomal RNA [rRNA], and other small RNAs) and associated proteins. tRNAs are vital for protein translation, decoding mRNA codons into amino acids via precise codon-anticodon interactions within the ribosome (Ogle et al, 2002). A key question in synthetic biology is determining the minimum number of tRNA isotypes required for cellular viability. The synthetic minimal genome organism *Mycoplasma mycoides* JCVI-syn3.0 uses 27 tRNA isotypes (Hutchison et al, 2016), while the naturally occurring endosymbiotic bacterium *Candidatus Nasuia deltocephalinicola*, with the smallest known bacterial genome, encodes 28 tRNA isotypes (Bennett and Moran, 2013). Intriguingly, the apicoplast, a secondary relict plastid organelle, of *Plasmodium falciparum* contains only 25 tRNA isotypes (Arisue et al, 2012; Aurrecoechea et al, 2008; Elahi and Prigge, 2023; Preiser et al, 1995; Wilson et al, 1996), aligning more closely with theoretical predictions for a minimal tRNA set (Alkatib et al, 2012). Both theoretical considerations and empirical data suggest that efficient mRNA codon decoding by a minimal tRNA set is facilitated by maximal use of wobble and superwobbling (where a tRNA with an unmodified U at the wobble position recognizes all four nucleotides at the third codon position) base pairing between the third mRNA codon base (wobble base) and the first tRNA anticodon base (position 34, wobble position), along with post-transcriptional tRNA modifications.

tRNAs undergo extensive post-transcriptional modifications across all kingdoms of life (Björk, 1994). These modifications, particularly at the wobble position of the anticodon, are crucial for fine-tuning wobble base pairing, either promoting or restricting specific interactions. In addition, modifications at other positions of the anticodon loop (positions 32–38) stabilize tRNA–codon interactions, enhancing translational accuracy. The presence of genes for enzymes involved in anticodon loop modifications, even in smaller bacterial genomes (580–1840 Kb) (Bennett and Moran, 2013; de Crécy-Lagard et al, 2012; Grosjean et al, 2014), further underscores the critical role of these modifications for living organisms.

To date, over 30 distinct wobble modifications have been characterized (Boccaletto et al, 2018). One such modification, lysidine (L or k$^2$C), is a lysine-containing derivative of cytidine (Fig. 1A) at the wobble position (C34) of minor isoleucine (Ile) tRNAs with CAU anticodon (tRNA$^{Ile}_{CAU}$) (Kuratani et al, 2007; Muramatsu et al, 1988b; Nakanishi et al, 2009; Nakanishi et al, 2005; Numata, 2015; Soma et al, 2003; Suzuki and Miyauchi, 2010).

[1]Department of Molecular Microbiology and Immunology, Johns Hopkins University, Baltimore, MD, USA. [2]The Johns Hopkins Malaria Research Institute, Baltimore, MD, USA.
[✉]E-mail: aelahi3@jhu.edu; sprigge2@jhu.edu

  

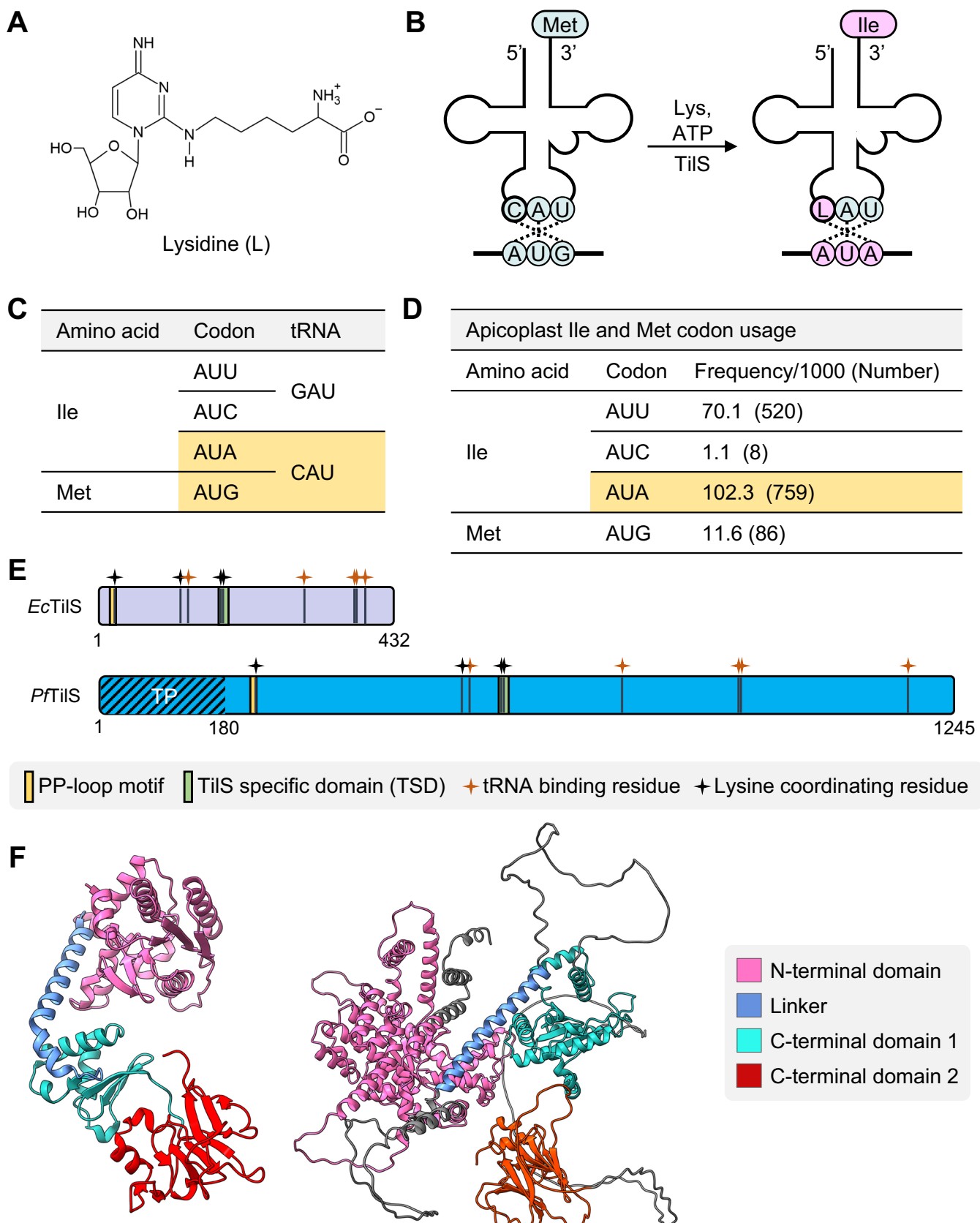

Figure legend:

| | PP-loop motif | | TilS specific domain (TSD) | ✦ tRNA binding residue | ✦ Lysine coordinating residue |

**Figure 1. *Plasmodium falciparum* contains a putative tRNA-isoleucine lysidine synthetase (TilS).**

(A) Chemical structure of lysidine (L). (B) The tRNA$_{CAU}$ binds the AUG codon and is charged with Methionine (Met), unless tRNA-isoleucine lysidine synthetase (TilS) modifies C34 to L34 using lysine (Lys) and ATP as substrates. The tRNA$_{CAU}$ with the LAU anticodon recognizes AUA codons, and is charged with isoleucine (Ile). (C) Two tRNAs in the apicoplast genome decode Met and Ile codons. tRNA$_{CAU}$ can potentially decode both AUA and AUG codons (highlighted in yellow). (D) Met and Ile codon usage in *Plasmodium* apicoplast-genome-coded proteins. Yellow highlight shows the most frequently used Ile codon. Refer to Fig. EV1 for codon usage of all proteins encoded by the apicoplast genome. (E) Residues important for tRNA binding (dark orange symbols) and lysine binding (black symbols) are mapped on *Escherichia coli* TilS (432 residues) and a putative 1245-residue *P. falciparum* TilS. TP, apicoplast targeting peptide. Refer to Appendix Fig. S1 for the multiple sequence alignment of TilS orthologs. (F) AlphaFold predicted structure of *Pf* TilS residues 181-1245 (right) shows a folding pattern similar to that of *E. coli* TilS (PDB id: 1NI5, left).

The nearly ubiquitous presence of the tRNA-isoleucine lysidine synthetase (TilS) enzyme, responsible for lysidine formation (Fig. 1B), strongly suggests the near-universal occurrence of this modification across bacterial species (Grosjean et al, 2014; Ikeuchi et al, 2005; Matsugi et al, 1996; Moriya et al, 1994; Muramatsu et al, 1988a; Muramatsu et al, 1988b). In its unmodified form, tRNA $^{Ile}_{CAU}$ behaves similarly to tRNA$^{Met}_{CAU}$, decoding the AUG codon as methionine (Met). However, the introduction of the L34 modification alters the codon specificity of tRNA$^{Ile}_{CAU}$, enabling it to be charged with Ile and decode the AUA codon as Ile. Consequently, this modification results in a shift in both codon and amino acid specificity. The essential role of TilS in accurate translation is evident from the translational defects observed in TilS-deficient *Escherichia coli* (Soma et al, 2003). Recognizing its importance, the *tilS* gene was included in the minimal synthetic genome *M. mycoides* JCVI-syn3.0 (Hutchison et al, 2016), as well as in all proposed minimal genome concepts to date (Forster and Church, 2006; Garzón et al, 2022; Gil and Peretó, 2015; Gil et al, 2004; Grosjean et al, 2014).

The complete set of 25 tRNA isotypes in the apicoplast genome of the malaria parasite *P. falciparum* represents, to the best of our knowledge, the minimal tRNA set observed to date, and also aligns with the theoretically proposed minimum tRNA number required for cell viability (Alkatib et al, 2012). This makes *P. falciparum* an intriguing model for studying minimal translational machinery. Although the apicoplast encodes all its rRNAs and tRNAs in its genome, it has transferred most of its protein-coding genes, including those for tRNA modification, to the nuclear genome. These proteins are then trafficked back to the organelle via an N-terminal bipartite transit peptide (Waller et al, 2000). In this work, we identified a TilS ortholog (*Pf*TilS) encoded in the nuclear genome and showed that it is trafficked to the apicoplast organelle. Using an apicoplast metabolic bypass system (Swift et al, 2020b), we showed that *Pf*TilS is essential for apicoplast maintenance and parasite viability. Successful complementation of *Pf*TilS with a characterized bacterial TilS enzyme allowed us to conclude that *Pf*TilS is essential due to its lysidine synthesis activity. To our knowledge, this is the first characterization of TilS in a eukaryote, significantly impacting future eukaryotic organelle research and strengthening our understanding of minimal translational machinery.

## Results

### Apicomplexan parasite contains a TilS ortholog

Among the 25 apicoplast genome encoded tRNA isotypes, Met and Ile decoding is accomplished with three tRNA$_{CAU}$ (currently annotated as tRNA$^{Met}_{CAU}$) and one tRNA$_{GAU}$ (annotated as tRNA$^{Ile}_{GAU}$) (Fig. 1C), respectively (Aurrecoechea et al, 2008; Preiser et al, 1995; Wilson et al, 1996). The tRNA$^{Ile}_{GAU}$ decodes AUU and AUC codons to Ile and tRNA$^{Met}_{CAU}$ decodes AUG as Met. However, there are no annotated tRNA$^{Ile}$ genes to decode AUA codons. In absence of direct evidence of tRNA import into secondary plastid-like organelles, it was proposed that tRNA$^{Met}_{CAU}$ can decode AUA codon for Ile with the L34 tRNA modification at the wobble position (Preiser et al, 1995). Intriguingly, AUA is the most frequently used codon for Ile for the 30 proteins encoded in the apicoplast genome (Fig. 1D, Fig. EV1). A recent investigation identified 28 distinct tRNA modifications within the total tRNA pool of *P. falciparum* (Ng et al, 2018). The study did not identify any apicoplast-specific modifications and the authors attributed this to the relatively low contribution of apicoplast RNA (0.5–2% rRNA compared to nuclear-encoded rRNAs) (Ng et al, 2018). It is likely, however, that modifications like lysidine are made in the apicoplast due to the prokaryotic origin (Fast et al, 2001; Janouskovec et al, 2010; Moore et al, 2008) of the apicoplast and the predominance of AUA codons in the apicoplast genome. We identified a putative TilS enzyme (PF3D7_0411200) with 28% sequence identity to *E. coli* TilS (*Ec*TilS). This protein is currently annotated as a putative PP-loop family protein (Aurrecoechea et al, 2008) and bioinformatic analysis predicts this nuclear-encoded protein to be trafficked to the apicoplast (Foth et al, 2003). Hereafter, we named this putative protein as *P. falciparum* (*Pf*) TilS.

Multiple sequence alignment (MSA) of the putative *Pf*TilS with TilS orthologs from diverse organisms, including *Synechocystis* sp. (*Sy*TilS), *Aquifex aeolicus* (*Aa*TilS), *Mycoplasma genitalium* (*Mg*TilS), *Ec*TilS, *Geobacillus kaustophilus* (*Gk*TilS), and *Arabidopsis thaliana* (*At*RSY3), revealed the presence of an N-terminal extension of 180 amino acid (aa) residues in *Pf*TilS (Fig. 1E; Appendix Fig. S1), which is predicted to function as a bipartite transit peptide directing the protein through the secretory system to the apicoplast (Foth et al, 2003). *Pf*TilS also exhibits conservation in functional domains critical for TilS activity. The protein retains conserved residues responsible for tRNA binding, lysine coordination, and a highly conserved ATP-binding PP-loop motif (SGGXDS) (Fig. 1E; Appendix Fig. S1) (Nakanishi et al, 2005; Soma et al, 2003). In addition, *Pf*TilS harbors the TilS-specific domain (TSD) where lysine interacts with the enzyme via hydrophobic interaction (Nakanishi et al, 2005). The PP-loop motif plays a crucial role in TilS function, activating the C2 position of the target tRNA's cytidine 34 to form an adenylate intermediate in an ATP-dependent manner. Subsequently, the ε-amino group of lysine performs a nucleophilic attack on the C2 position of the adenylate intermediate, resulting in lysidine formation (Ikeuchi et al, 2005; Kuratani et al, 2007; Nakanishi et al, 2009; Nakanishi

et al, 2005; Numata, 2015). Other pathogenic apicoplast-containing apicomplexans also seem to possess nuclear-genome-encoded TilS orthologs with significant conservation in functional domains and important residues (Appendix Fig. S2). These observations suggest conservation of AUA decoding in apicoplast-containing apicomplexans. Similar to a type-I TilS protein *Ec*TilS, AlphaFold structural prediction (Jumper et al, 2021; Varadi et al, 2021) suggests that *Pf*TilS adopts a comparable folding architecture with three distinct domains: an N-terminal domain (NTD) and two C-terminal domains (CTD1 and CTD2) (Fig. 1F). Despite similar predicted architecture for *Pf*TilS and *Ec*TilS, maximum likelihood phylogenetic analysis (Jones et al, 1992) of TilS orthologs shows a relatively distant evolutionary relationship between *P. falciparum* and *E. coli* orthologs (Appendix Fig. S3).

## *Plasmodium falciparum* TilS localizes to the apicoplast

To experimentally validate the predicted apicoplast localization of *Pf*TilS, we generated two parasite lines. One line expressed a full-length codon-modified *Pf*TilS (residues 1–1245) fused to three tandem C-terminal V5 epitope tags (designated *pftilS*[+]). The other line expressed a codon-modified N-terminally truncated *Pf*TilS (residues 181–1245) fused to a similar V5 tag (designated *trpftilS*[+]) (Fig. 2A). Both constructs were integrated into the genome of PfMev[attB] parasites (Swift et al, 2021) using a knock-in approach mediated by mycobacteriophage integrase-based recombination (Spalding et al, 2010) (Fig. 2B). Successful generation of both *pftilS*[+] and *trpftilS*[+] lines was confirmed by PCR (Fig. 2C).

Immunofluorescence analysis of *pftilS*[+] parasites revealed colocalization of V5-tagged *Pf*TilS with the apicoplast marker protein, acyl carrier protein (ACP) (Waller et al, 2000) (Fig. 2D). This colocalization validates the predicted apicoplast localization of *Pf*TilS and aligns with previous observations of *Pf*TilS protein within the apicoplast proteome identified by proximity biotinylation-based proteomics (Boucher et al, 2018). Conversely, truncated *Pf*TilS in parasites (*trpftilS*[+]) displayed a diffuse cytosolic localization pattern, as expected for a protein lacking an apicoplast targeting peptide. This localization pattern demonstrates that the N-terminal 180 aa residues of *Pf*TilS are required for apicoplast trafficking.

## *Pf*TilS is essential for parasite survival and apicoplast maintenance

We used CRISPR/Cas9-mediated gene deletion (Ghorbal et al, 2014) to assess *Pf*TilS essentiality in a metabolic bypass parasite line, PfMev[attB] (Fig. 3A). PfMev[attB] parasites contain an engineered metabolic pathway to synthesize the isoprenoid precursors IPP (isopentenyl pyrophosphate) and DMAPP (dimethylallyl pyrophosphate) from mevalonate, allowing parasites to survive loss of the apicoplast organelle and its endogenous isoprenoid pathway (Swift et al, 2020b; Swift et al, 2021). In addition, these parasites also express apicoplast-localized superfolder green (api-SFG) (Roberts et al, 2019) allowing fluorescent visualization of the organelle (Swift et al, 2020b; Swift et al, 2021).

We confirmed the successful deletion of *pftilS* using genotyping PCR (Fig. 3B). We previously demonstrated that deletion of certain apicoplast tRNA modification genes leads to apicoplast disruption and loss of the organellar genome (Swift et al, 2023). To investigate

whether *pftilS* deletion resulted in apicoplast genome loss, we attempted to amplify the *sufB* gene from the apicoplast genome of Δ*pftilS* parasites. In Δ*pftilS* parasites, our attempt to amplify *sufB* was unsuccessful (Fig. 3C), suggesting apicoplast genome loss. Consistent with this finding, Δ*pftilS* parasites displayed the characteristic phenotype of apicoplast organelle disruption—multiple, discrete api-SFG labeled vesicles (Yeh and DeRisi, 2011) (Fig. 3D). As anticipated for parasites with disrupted apicoplasts, Δ*pftilS* parasites exhibited a strict dependence on mevalonate for survival (Fig. 3E).

Collectively, these results demonstrate that *Pf*TilS is essential for both parasite survival and apicoplast maintenance. Given the predicted role of *Pf*TilS in catalyzing a crucial tRNA modification, its deletion likely disrupts the efficient translation of essential apicoplast-encoded proteins, ultimately leading to apicoplast loss and parasite death.

## *Escherichia coli* TilS can be expressed in the *P. falciparum* apicoplast

*Ec*TilS is the most extensively studied and well-characterized TilS ortholog to date. Partial inactivation of *Ec*TilS led to a decrease in level of the L34 modification and to decoding defects for AUA codons (Soma et al, 2003). In addition, detailed biochemical (Ikeuchi et al, 2005; Numata, 2015; Soma et al, 2003) and structural (Nakanishi et al, 2005; Numata, 2015) studies of *Ec*TilS have been reported. To elucidate the role of *Pf*TilS, we attempted to complement its activity with *Ec*TilS. We achieved this by integrating the *ectilS* gene into PfMev[attB] parasites using mycobacteriophage integrase-mediated recombination (Spalding et al, 2010) (Fig. 4A). Hereafter, we refer to this parasite line as *ectilS*[+]. The *ectilS* expression cassette encodes a conditional localization domain (CLD) at the N-terminus to allow inducible control over protein localization (Roberts et al, 2019). It also incorporates an mCherry tag on the C-terminus for visualization by live-cell fluorescence. In addition, an aptamer array was included in the 3' untranslated region (UTR) of the gene. The CLD facilitates conditional mislocalization of the tagged protein (Roberts et al, 2019), while the TetR-DOZI system regulates protein expression by controlling mRNA stability (Ganesan et al, 2016; Rajaram et al, 2020) (Fig. 4B).

We confirmed successful generation of the *ectilS*[+] knock-in line with genotyping PCR (Fig. 4C) and expression of *Ec*TilS-mCherry fusion protein was confirmed via immunoblot (Fig. 4D; Appendix Fig. S4). These parasites did not display significant growth difference compared to the parental line (p = 0.3095, two-tailed Mann–Whitney U test), suggesting that *Ec*TilS over expression is not toxic to parasites (Fig. 4E). Live epifluorescence microscopy revealed apicoplast localization of *Ec*TilS protein in the *ectilS*[+] line, evidenced by colocalization of mCherry fluorescence with that of the apicoplast marker api-SFG (Fig. 4F). Collectively, these results demonstrate the successful expression and apicoplast targeting of *Ec*TilS in *P. falciparum*.

## *Escherichia coli* TilS functionally complements *P. falciparum* TilS activity in the apicoplast

To assess whether *Ec*TilS could functionally complement *Pf*TilS, we attempted to delete the endogenous parasite *pftilS* gene in the *ectilS*[+] parasite line using the same guide RNA and repair plasmids

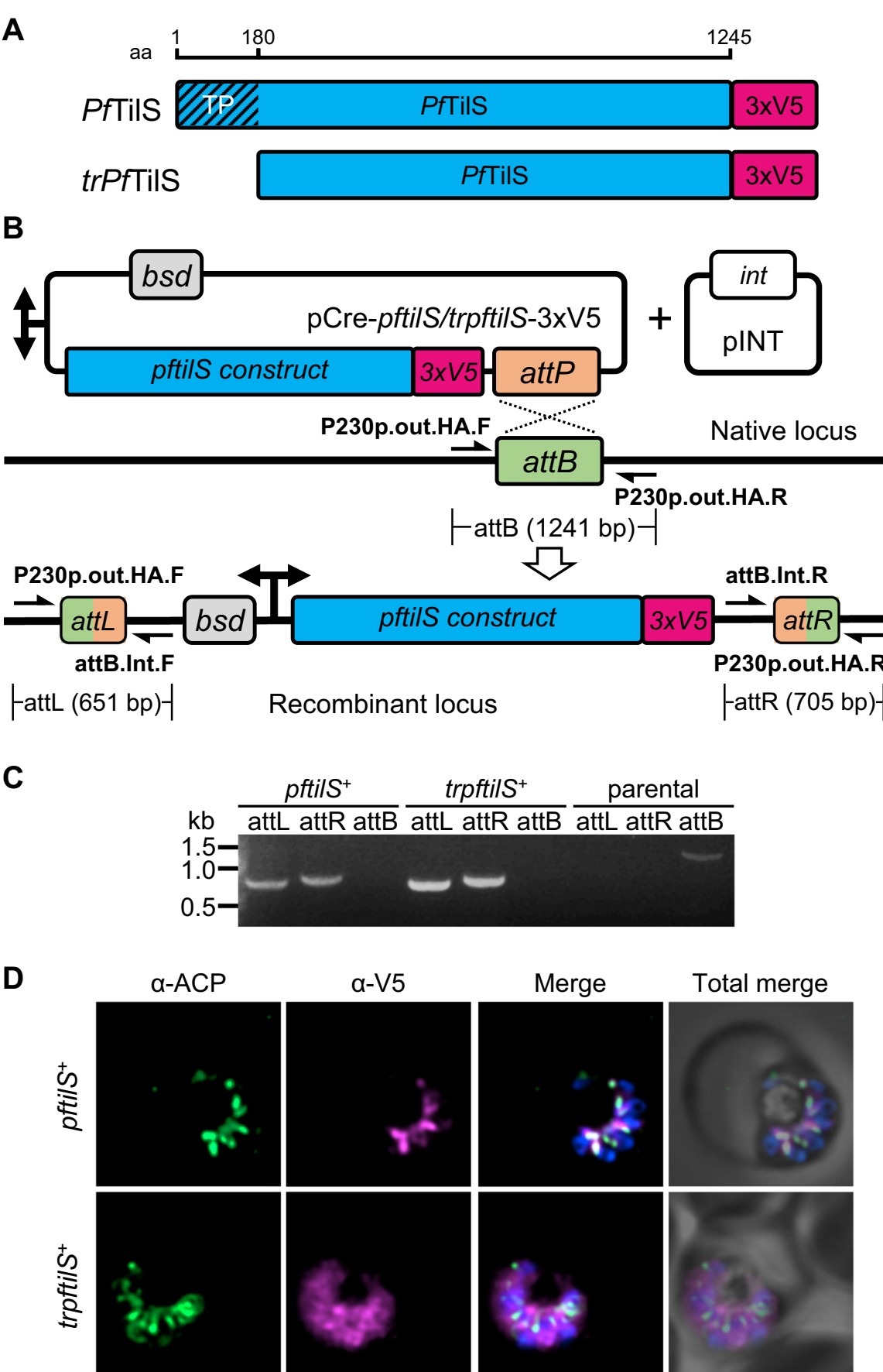

◄  **Figure 2.   *Plasmodium falciparum* TilS localizes to the apicoplast.**

(A) Scheme showing synthetic *Pf*TilS constructs. The scale at the top shows the protein length (in amino acid residues, aa) of full-length *Pf*TilS and a truncated construct (tr*Pf*TilS) lacking the predicted trafficking peptide (TP). (B) Schematic illustration (not to scale) of pCre plasmid insertion into the attB locus of PfMev$^{attB}$ parasites to generate *pftilS$^{+}$* or *trpftilS$^{+}$* parasites. A bidirectional promoter drives the expression of blasticidin-S-deaminase (*bsd*) and the *Pf*TilS construct (*pftils-3xV5* or *trpftilS-3xV5*). Plasmid pINT expresses integrase (*int*) for catalyzing attB/attP recombination. Primer pairs (black half arrow) for PCR amplification of the attB region of the parental line and the recombinant attL and attR regions with expected amplicon sizes are depicted. Refer to Appendix Table S1 for primer sequences. (C) PCR amplification of attL and attR regions confirms plasmid integration in both *pftilS$^{+}$* and *trpftilS$^{+}$* parasites. Amplification of the attB region from the PfMev$^{attB}$ parasite line (parental) was used as a control. Expected amplicon size with corresponding primer pairs are depicted in (B). DNA markers in kilobases (kb). (D) Immunofluorescence microscopy demonstrates colocalization of full-length *Pf*TilS (*pftilS$^{+}$*) tagged with a C-terminal 3xV5 epitope (magenta) with the apicoplast marker ACP (acyl carrier protein, green). Truncated *Pf*TilS (*trpftilS$^{+}$*) exhibits a diffuse cytosolic distribution (magenta) in the bottom panels. DAPI staining (blue) marks nuclear DNA in both panels. Images represent a field of 10 µm × 10 µm. Source data are available online for this figure.

used in Fig. 3. We hypothesized that successful complementation of *Pf*TilS activity by *Ec*TilS would result in a phenotype similar to parental PfMev$^{attB}$ parasites, with intact apicoplasts and no requirement for mevalonate for growth (Fig. 5A). In repeated independent transfections, we successfully generated *ectilS$^{+}$ΔpftilS* parasite lines (Fig. 5B). These parasites grew without mevalonate supplementation (Fig. 5C) and appeared to have intact apicoplasts as assessed by both live-cell imaging (Fig. 5D) and PCR amplification of the apicoplast-encoded *sufB* gene (Fig. 5E). Furthermore, these parasites did not show any growth defect compared to parental PfMev$^{attB}$ parasites ($p = 0.8413$, two-tailed Mann–Whitney U test) (Fig. 5F). These findings strongly suggest that *Ec*TilS can complement the function of *Pf*TilS.

To further solidify the evidence for complementation by *Ec*TilS, we next aimed to conditionally knockdown *Ec*TilS expression in the *ectilS$^{+}$ΔpftilS* parasite line. This was achieved by using the TetR-DOZI and CLD systems previously described in Fig. 4B. The TetR-DOZI system allows for inducible control of protein expression by regulating mRNA stability in response to non-toxic small molecule anhydrous tetracycline (aTc) (Ganesan et al, 2016; Rajaram et al, 2020). The CLD directs the tagged *Ec*TilS protein to the apicoplast under normal conditions. However, upon addition of the non-toxic small molecule ligand *Shield1*, the tagged protein is redirected to the parasitophorous vacuole (Fig. 5G) (Roberts et al, 2019). We monitored the growth of *ectilS$^{+}$ΔpftilS* parasites under permissive (aTc present, *Shield1* absent) and non-permissive conditions (aTc absent, *Shield1* present) for 8 days. Under the non-permissive condition, the *ectilS$^{+}$ΔpftilS* parasites displayed a significant growth defect from day 4 ($p = 0.0011$, two-way ANOVA, Sidak-Bonferroni method) (Fig. 5H). Live epifluorescence microscopy on day 4 revealed a disrupted apicoplast phenotype in parasites grown under non-permissive conditions (Fig. 5I). These results demonstrate that *Ec*TilS function is indispensable for maintaining apicoplast integrity following *Pf*TilS deletion and highlight the importance of TilS activity in apicoplast function and parasite survival.

## Discussion

The conservation of enzymes that modify tRNA anticodon loop nucleotides (positions 32–38) in nearly all bacteria underscores the critical role that these modifications play in ensuring accurate and efficient protein translation in prokaryotes. Comparative genomic analyses of organisms with reduced genomes consistently identify tRNA modifying enzymes targeting the anticodon loop as essential components of minimal translational systems (Forster and Church,

2006; Garzón et al, 2022; Gil and Peretó, 2015; Gil et al, 2004; Grosjean et al, 2014; Hutchison et al, 2016). TilS, the only known enzyme catalyzing lysidine modification at the anticodon wobble position of tRNA$^{Ile}_{CAU}$, exemplifies this. Lysidine enables precise AUA codon recognition for isoleucine incorporation while preventing AUG codon misreading, thereby dictating both codon specificity and amino acid assignment for tRNA$^{Ile}_{CAU}$. In this work, we took advantage of unique features of apicoplast biology to characterize the first eukaryotic TilS enzyme and show that it is essential for apicoplast maintenance and parasite survival.

The apicoplast (a relict secondary plastid) of the apicomplexan parasite *P. falciparum* has an endosymbiotic origin and encodes a minimal set of 25 complete tRNA isotypes within its circular genome (Arisue et al, 2012; Aurrecoechea et al, 2008; Elahi and Prigge, 2023; Preiser et al, 1995; Wilson et al, 1996). This number is lower than the synthetic minimal genome organism *M. mycoides* JCVI-syn3.0, which contains 27 tRNA isotypes (Hutchison et al, 2016), positioning the apicoplast as a natural model for studying minimal translational machinery. Another attractive feature of the apicoplast is the ability to delete essential proteins in a metabolic bypass parasite line (PfMev$^{attB}$) that allows the parasites to produce essential isoprenoid products in the cytosol without relying on apicoplast metabolism (Swift et al, 2020b). In this study, we identified a potential TilS enzyme (*Pf*TilS) encoded in the nuclear genome and used mevalonate supplementation in the PfMev$^{attB}$ line to show that *Pf*TilS is essential for parasite survival (Fig. 3). The *ΔpftilS* line displayed a 'disrupted apicoplast' phenotype, including loss of the apicoplast genome and the appearance of multiple vesicles containing nucleus-encoded apicoplast proteins. This phenotype was observed in PfMev$^{attB}$ parasites when proteins required for apicoplast maintenance were deleted (Swift et al, 2023; Swift et al, 2020a). Importantly, this was the phenotype observed when the only essential apicoplast tRNA modifying enzyme described (MnmA) to date was deleted (Swift et al, 2023).

We used a complementation approach to establish the essential function of *Pf*TilS. Biochemical and genetic analyses show that the TilS ortholog from *E. coli* (*Ec*TilS) catalyzes the formation of lysidine on tRNA$^{Ile}_{CAU}$ (Ikeuchi et al, 2005; Numata, 2015; Soma et al, 2003). We found that *Ec*TilS successfully complemented the loss of *Pf*TilS, resulting in parasites with intact apicoplasts. Subsequent knockdown of the complemented *Ec*TilS caused apicoplast disruption, highlighting the essentiality of TilS lysidiny-lation activity for both apicoplast maintenance and parasite survival (Fig. 5). These findings strongly suggest that *Plasmodium* TilS catalyzes the lysidinylation of tRNA$_{CAU}$ within the apicoplast (Fig. 6A). Disruption of tRNA lysidinylation likely disrupts the

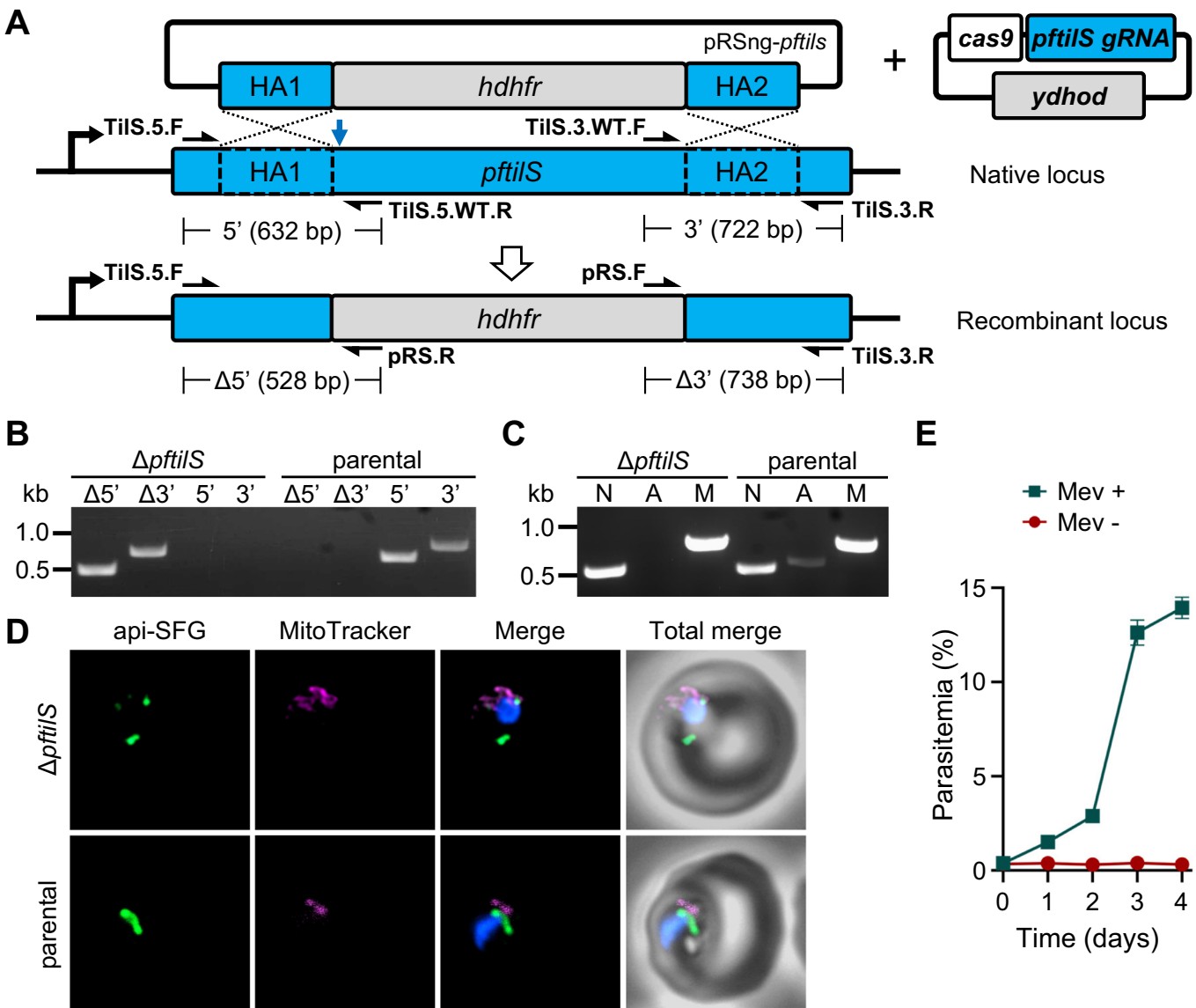

**Figure 3. *Pf*TilS is essential for apicoplast maintenance and parasite survival.**

(A) Schematic illustrating double-crossover homologous recombination for *pftilS* gene knockout. A repair plasmid (pRSng-*pftilS*) harboring two homology arms (HAs) flanks the desired modification site within the native *pftilS* locus. Cas9 endonuclease with guide RNA (Cas9-*pftilS*gRNA plasmid) introduces a double-stranded break (blue arrow) in the native locus, facilitating homologous recombination with the repair plasmid leading to the recombinant Δ*pftilS* locus. Primer positions and directions (black half arrows) for confirming gene knockout are indicated. Refer to Appendix Table S1 for primer sequences. *hdhfr*, human dihydrofolate reductase; *ydhod*, yeast dihydroorotate dehydrogenase. (B) Genotyping PCR verifies *pftilS* deletion in Δ*pftilS* parasites, shown by the presence of amplicons for the Δ5′ and Δ3′ loci at the integration site, but not for the native loci (5′ and 3′) found in the PfMev$^{attB}$ (parental) parasites. The primers and expected amplicon sizes are depicted in (A). (C) Attempted PCR amplification of *ldh*, *sufB*, and *cox1* genes of the parasite nuclear (N), apicoplast (A), and mitochondrial (M) genomes, respectively, in Δ*pftilS* and PfMev$^{attB}$ (parental) parasites. Lack of an amplicon for *sufB* in the Δ*pftilS* parasites indicates the loss of the apicoplast genome. Refer to Appendix Table S1 for primer names and sequences and Methods section for expected amplicon sizes. (D) Representative epifluorescence microscopy images of Δ*pftilS* parasites shows multiple discrete vesicles (top panel) demonstrating a disrupted apicoplast, compared to an intact apicoplast in PfMev$^{attB}$ parasites (parental, bottom panel). Api-SFG protein (green) marks the apicoplast, the mitochondrion is stained with MitoTracker (magenta), and nuclear DNA is stained with DAPI (blue). Each image depicts a field of 10 μm × 10 μm. (E) The Δ*pftilS* parasites are dependent on mevalonate (Mev) for growth. Asynchronous parasites were grown with or without 50 μM Mev and parasitemia was monitored every 24 h by flow cytometry for four days. Data points represent daily mean parasitemia ± standard error of mean (SEM) from two biological replicates, each with four technical replicates. In (B) and (C), DNA markers are in kilobases (kb). Source data are available online for this figure.

proper decoding of the most frequently used isoleucine codon (AUA) in apicoplast-genome-encoded proteins. Consequently, this would hinder the translation of essential apicoplast proteins, ultimately leading to organelle dysfunction and loss (Fig. 6B).

Similar to bacteria, archaea employ a different cytidine derivative, agmatinylcytidine (agm²C), at position C34 of tRNA$_{CAU}$ to decode the AUA codon as isoleucine while avoiding misreading of AUG (Ikeuchi et al, 2010; Köhrer et al, 2008; Mandal et al, 2010).

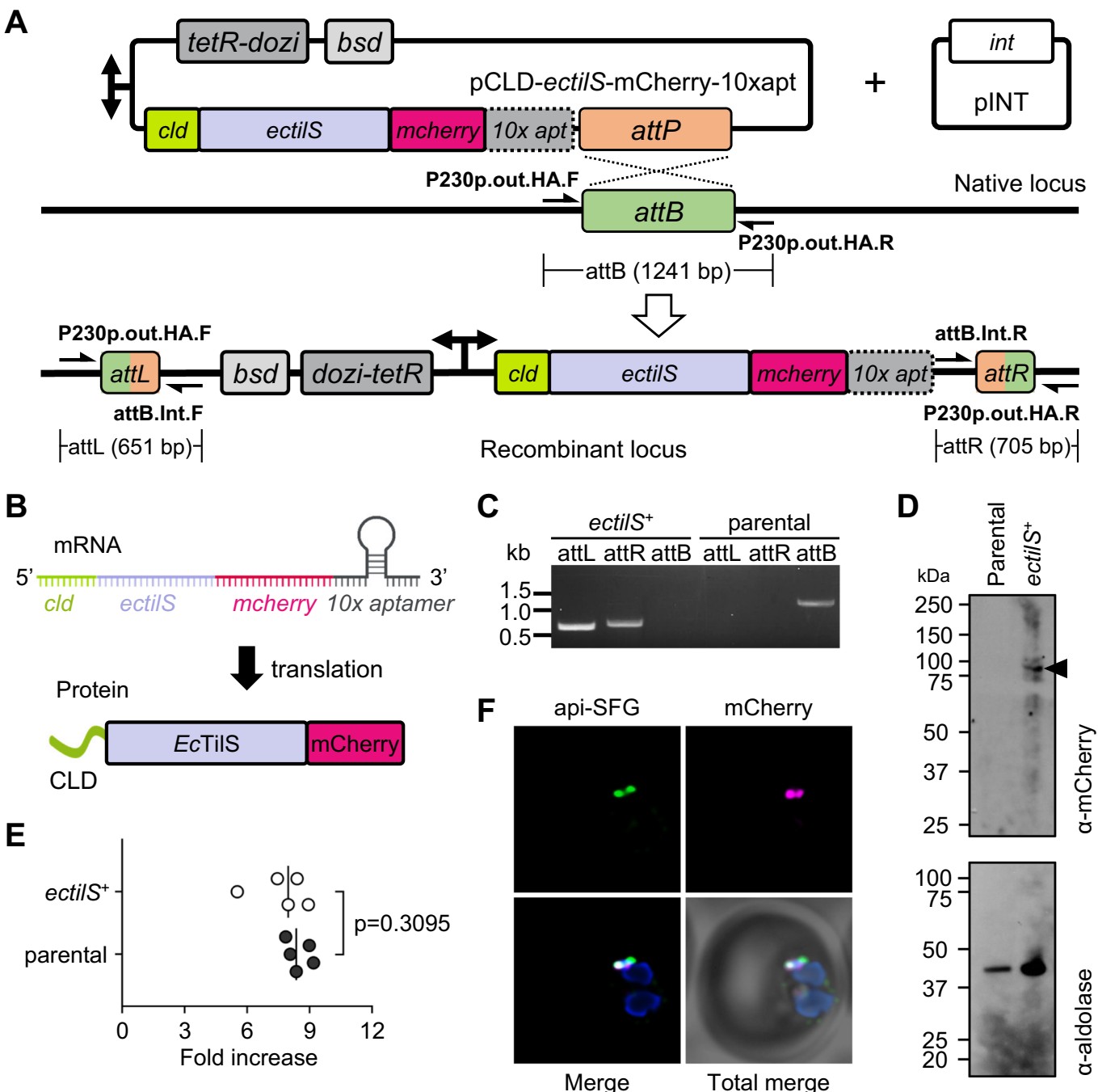

**A**

**B** mRNA
5' — cld — ectilS — mcherry — 10x aptamer — 3'

↓ translation

Protein
CLD — EcTilS — mCherry

**C** ectilS⁺ | parental
kb | attL attR attB | attL attR attB

**D** Parental | ectilS⁺
α-mCherry
α-aldolase

**E**
ectilS⁺
parental
p=0.3095
Fold increase

**F** api-SFG | mCherry
Merge | Total merge

This essential modification is catalyzed by tRNA^Ile-agm²C synthetase (TiaS) in an ATP-dependent manner (Ikeuchi et al, 2010). Despite the chemical similarity between lysidine and agm²C, the catalytic domains of bacterial TilS and archaeal TiaS are markedly different, employing distinct catalytic mechanisms (Ikeuchi et al, 2005; Muramatsu et al, 1988a; Muramatsu et al, 1988b; Nakanishi et al, 2005; Osawa et al, 2011; Soma et al, 2003; Suzuki and Miyauchi, 2010; Suzuki and Numata, 2014; Terasaka et al, 2011). Sequence alignment of *P. falciparum* TilS with *Archaeoglobus fulgidus* TiaS reveals minimal to no conservation of catalytic residues or domains (Appendix Fig. S5). Combined with

experimental complementation of *P. falciparum* TilS using bacterial TilS, these results strongly support the lysidinylation activity of *P. falciparum* TilS.

Although our complementation assays offer robust support for the lysidinylation activity of *Pf*TilS, the lysidine modification in apicoplast-encoded tRNA_CAU has not yet been directly observed. Unfortunately, apicoplast disruption and subsequent loss of its genome upon TilS depletion (knockout or knockdown) preclude the investigation of any ribonucleoside modifications within apicoplast tRNAs. In addition, a recent study surveying the epitranscriptome of *P. falciparum* did not detect ribonucleoside

◀ **Figure 4.** *Escherichia coli* **TilS can be expressed in the *P. falciparum* apicoplast.**

(A) Schematic illustration of pCLD-*ectilS*-mCherry-10xapt plasmid insertion into the attB locus of PfMev^attB parasites to generate the *ectilS*^+ line. The *tetR-dozi* inducible system regulator and blasticidin-*S*-deaminase (*bsd*) are separated by a T2A viral skip peptide (not shown). The *cld-ectilS-mcherry-10xapt* and *tetR-dozi-2A-bsd* cassettes are expressed under a single bidirectional promoter. This plasmid was co-transfected with the pINT plasmid, which expresses integrase (*int*) for catalyzing attB/attP recombination. Primer pairs (black half arrows) for PCR amplification of the attB region of the parental line and the recombinant attL and attR regions are marked along with the expected amplicon sizes. Refer to Appendix Table S1 for primer sequences. cld, conditional localization domain. (B) Expected transcript and protein in *ectilS*^+. (C) PCR amplification of attL and attR regions confirms plasmid integration in *ectilS*^+ parasites. Amplification of the attB region from the PfMev^attB parasite line (parental) was used as a control. Expected amplicon size with corresponding primer pairs are depicted in (A). DNA markers are in kilobases (kb). (D) Immunoblot of saponin-isolated PfMev^attB (parental) and *ectilS*^+ parasite lysates with anti-mCherry antibody (top panel) confirms expression of *Ec*TilS-mCherry fusion protein (expected molecular weight 91 kDa, black arrowhead). Anti-aldolase immunoblot shows relative loading levels (bottom panel). Protein markers are in kilodaltons (kDa). Refer to Appendix Fig. S4 for the uncropped blot image. (E) Growth comparison of *ectilS*^+ (white circles) and the PfMev^attB parental line (dark gray circles) with 0.5 μM aTc supplementation shows no significant growth difference (two-tailed Mann–Whitney U test). Asynchronous parasites were cultured for a complete growth cycle (~48 h), after which parasitemia was determined. The ratio of the final and initial parasitemia (fold increase) represents the growth rate. Data are from five biological replicates, each with technical duplicates; *bars*, median fold increase. (F) Representative epifluorescence microscopy images of *ectilS*^+ parasites confirm the colocalization of *Ec*TilS-mCherry fusion protein (magenta) with the apicoplast api-SFG marker (green). The nuclear DNA is stained with DAPI (blue). Each image depicts a field of 10 μm × 10 μm. Source data are available online for this figure.

modifications from apicoplast tRNAs. This is likely due to the minor contribution of apicoplast tRNAs to the total parasite tRNA pool, potentially falling below the detection threshold of the employed methodology (Ng et al, 2018). Future studies using selective enrichment techniques to isolate apicoplast tRNAs, coupled with the inclusion of synthetic standards, offer a promising approach to directly observe the lysidine modification in apicoplast-encoded tRNA_{CAU}. Interestingly, the identification of TilS orthologs in chloroplast genomes (de Koning and Keeling, 2006) or in plant nuclear genomes with predicted chloroplast localization (Fages-Lartaud and Hohmann-Marriott, 2022; The Arabidopsis Genome Initiative, 2000) suggests a conserved mechanism for isoleucine decoding within diverse plastid organelles. While a recent study identified lysidine modifications in *A. thaliana* (Miyauchi et al, 2024), TilS enzymes in eukaryotic organelles remain uncharacterized (Gołębiewska et al, 2024).

The *P. falciparum* apicoplast genome contains three CAU anticodon-bearing tRNAs and all are annotated as tRNA^Met_{CAU}. This seems to be the case for other major apicoplast containing pathogenic apicomplexans also (Arisue et al, 2012; Aurrecoechea et al, 2008; Berná et al, 2021; de Koning and Keeling, 2006; Gardner et al, 2005; Garg et al, 2014; Kissinger et al, 2003; Reid et al, 2014). This raises the possibility that one of the three apicoplast-encoded tRNA^Met_{CAU} isoacceptors in *P. falciparum* is misannotated. In the archaeon *Haloarcula marismortui*, both bioinformatic and biochemical data demonstrated the misannotation of a CAU anticodon-bearing tRNA as tRNA^Met_{CAU}, when it was actually aminoacylated by isoleucyl-tRNA synthetases (Köhrer et al, 2008). To identify the potential tRNA substrate for *Pf*TilS, we conducted a maximum likelihood phylogenetic analysis (Jones et al, 1992) encompassing 83 tRNA genes with anticodons for methionine (CAU) and isoleucine (CAU, AAU, UAU, GAU) from archaea, eubacteria, higher eukaryotes, and apicomplexan apicoplast genomes. This analysis revealed a potential shared ancestry between one of the three *P. falciparum* apicoplast tRNA genes harboring a CAU anticodon (PF3D7_API00600) and experimentally validated bacterial TilS tRNA substrate (Fig. EV2). Previous studies using footprinting and structural analyses in *E. coli* have identified U33, C34, and A37 in the anticodon loop, the G27 + U43 base pair in the anticodon stem, and the C4 + G69 and C5 + G68 base pairs in the acceptor stem of tRNA^Ile_{CAU} as crucial determinants for TilS binding (Fig. EV3) (Ikeuchi et al, 2005). Although

PF3D7_API00600 tRNA lacks the stem determinants, it possesses the conserved bases within the anticodon loop, making it the best candidate for an apicoplast tRNA^Ile_{CAU}.

The determinants present in the anticodon and acceptor stems are conserved in γ-proteobacteria but not in other eubacterial tRNAs, including *B. subtilis* (Ikeuchi et al, 2005). This suggests that TilS recognition and discrimination mechanisms between similar tRNAs might vary across organisms. Nonetheless, apicoplast localization of *Pf*TilS and its functional complementation by *Ec*TilS in the apicoplast strongly suggests that at least one of the three apicoplast tRNA_{CAU} isoacceptors is modified by lysidinylation. As demonstrated in bacteria (Muramatsu et al, 1988a), the lysidine modification at the wobble position of tRNA_{CAU} is likely to be sufficient to enable its recognition by isoleucyl-tRNA synthetase rather than methionyl-tRNA synthetase in the apicoplast.

Codon usage varies widely between organisms. In *E. coli*, isoleucine is predominantly encoded by AUU (30.5/1000 codons) and AUC (18.2/1000 codons), with the AUA codon being rarely used (3.7/1000 codons) (Nakamura et al, 2000). By contrast, proteins encoded by the apicoplast genome exhibit a strong preference for AUA, followed by AUU (Fig. 1D, Fig. EV1). This preference aligns with the AT-rich nucleotide composition (89.1%) found in apicoplast protein-coding genes, where codons ending in A or T are favored (Fig. EV1), a pattern observed in other AT-rich endosymbiont genomes (de Crécy-Lagard et al, 2012; Moran, 1996). Notably, many endosymbionts also preferentially use AUA for isoleucine decoding (de Crécy-Lagard et al, 2012). This shift in codon preference likely reflects the evolutionary adaptation of TilS to efficiently recognize the AUA codon, optimizing translation in these AT-rich environments.

The current study identifies TilS, a tRNA-modifying enzyme localized to the apicoplast in the apicomplexan parasite *P. falciparum*. We demonstrate that TilS activity is required for apicoplast maintenance and parasite survival, presumably due to its crucial role in isoleucine and methionine decoding. These findings establish TilS as a fundamental component of minimal protein translation machinery and inform synthetic biology efforts to design highly reduced biological systems. These results also provide new insights into apicoplast biology in apicomplexan parasites and unveil a novel vulnerability in several of the important pathogens found in this group of organisms. This vulnerability presents a potential avenue for therapeutic intervention in the fight against

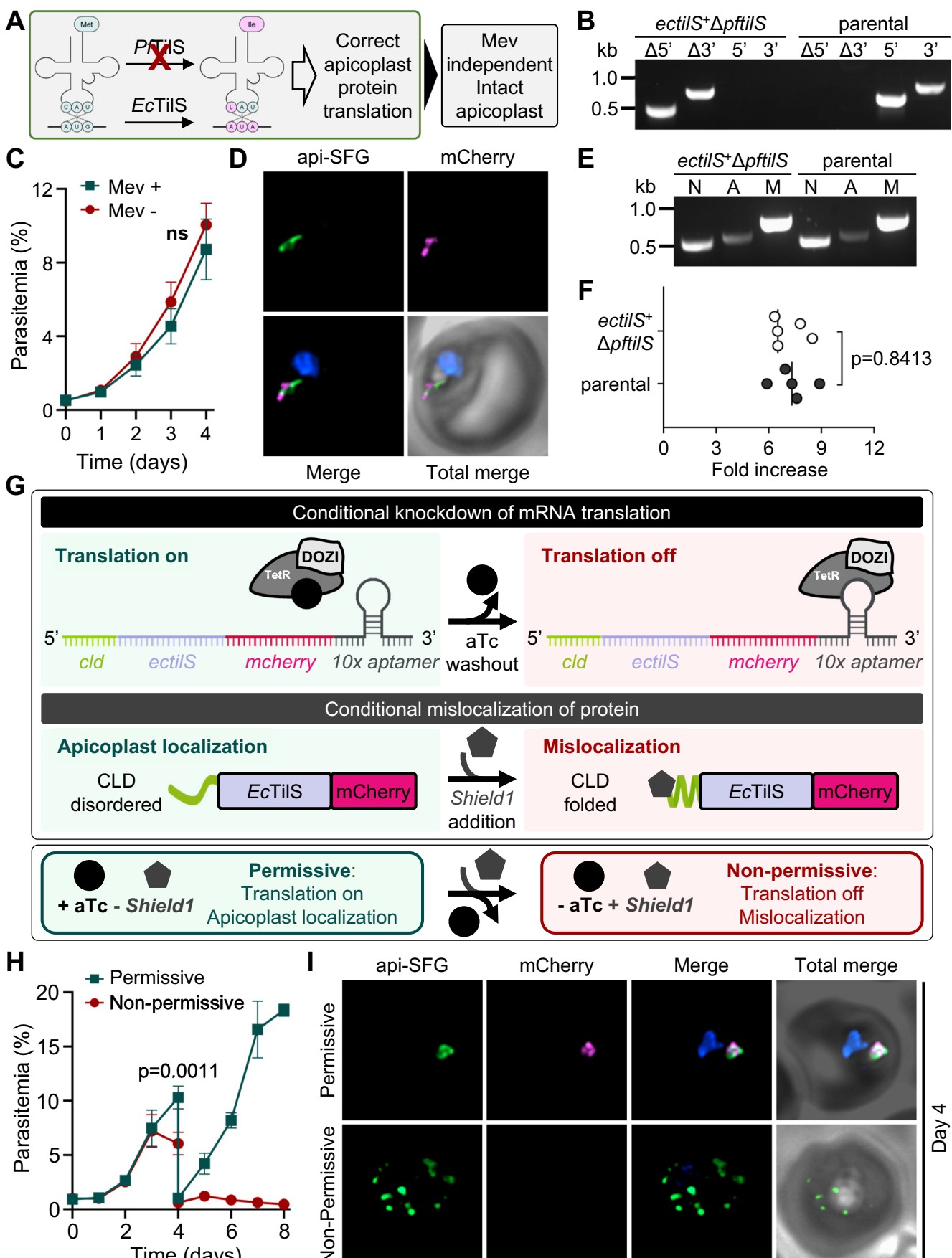

**Figure 5. Escherichia coli TilS complements the loss of P. falciparum TilS.**

(A) The anticipated outcome of *pftilS* deletion in the *ectilS⁺* parasite line is depicted in the schematic. *E. coli* TilS complementation of parasite TilS lysidinylation activity is sufficient for proper apicoplast protein translation and maintenance. (B) Genotyping PCR verifies *pftilS* deletion in *ectilS⁺ΔpftilS* parasites, shown by the presence of amplicons for the Δ5′ and Δ3′ loci at the integration site, but not for the native loci (5′ and 3′) found in the *ectilS⁺* (parental) parasites. The primers and expected amplicon sizes are depicted in Fig. 3A. (C) Growth of *ectilS⁺ΔpftilS* parasites does not require mevalonate (Mev). Asynchronous parasites were grown with or without 50 µM Mev in media containing 0.5 µM aTc. Parasitemia was monitored every 24 h by flow cytometry for four days. Data points represent daily mean parasitemia ± SEM from two biological replicates, each with four technical replicates; n.s., non-significant, two-way ANOVA (Sidak-Bonferroni method), $p > 0.05$. (D) Representative epifluorescence microscopy images of *ectilS⁺ΔpftilS* parasites show an intact apicoplast. (E) PCR detection of *ldh*, *sufB*, and *cox1* genes of the parasite nuclear (N), apicoplast (A), and mitochondrial (M) genomes, respectively, in *ectilS⁺ΔpftilS* and *ectilS⁺* (parental) parasites. Successful amplification of *sufB* in *ectilS⁺ΔtilS* parasites indicates the presence of the apicoplast genome. (F) Growth comparison of *ectilS⁺ΔpftilS* (white circles) and the PfMevᵃᵗᵗᴮ parental line (dark gray circles) with 0.5 µM aTc supplementation shows no significant growth difference (two-tailed Mann–Whitney U test). Asynchronous parasites were cultured for a complete growth cycle (~48 h), after which parasitemia was determined. The ratio of the final and initial parasitemia (fold increase) represents the growth rate. Data are from five biological replicates, each with technical duplicates; bars, median fold increase. (G) Schematic depicts the inducible control of *Ec*TilS-mCherry fusion protein in *ectilS⁺ΔpftilS* parasites. The TetR-DOZI system governs protein expression by regulating mRNA stability in response to anhydrous tetracycline (aTc). The conditional localization domain (CLD) directs the tagged *Ec*TilS protein to the parasitophorous vacuole upon *Shield1* administration; otherwise, it localizes to the apicoplast. This combined approach using aTc and *Shield1* offers enhanced control over *Ec*TilS-mCherry fusion protein levels. (H) Asynchronous *ectilS⁺ΔpftilS* parasites grew poorly (two-way ANOVA, Sidak-Bonferroni method) under non-permissive conditions (absence of 0.5 µM aTc and presence of 0.5 µM *Shield1*) compared to parasites grown under permissive conditions (presence of 0.5 µM aTc and absence of 0.5 µM *Shield1*). Parasitemia was monitored daily for eight days using flow cytometry. To prevent overgrowth, parasite cultures were diluted 1:10 on day 4. Data points represent the daily mean parasitemia ± SEM from two biological replicates, each with four technical replicates. (I) Representative epifluorescence microscopy of *ectilS⁺ΔpftilS* parasites on day 4 (from panel H) shows an intact apicoplast under permissive conditions. By contrast, multiple discrete vesicles suggesting a disrupted apicoplast were observed under non-permissive conditions. *Ec*TilS-mCherry fluorescence (magenta) is only visible under permissive conditions. In (D) and (I), The *Ec*TilS protein is tagged with mCherry (magenta), api-SFG protein (green) labels the apicoplast, and nuclear DNA is stained with DAPI (blue). Each image depicts a field of 10 µm × 10 µm. In (B) and (E), DNA markers are in kilobases (kb). Source data are available online for this figure.

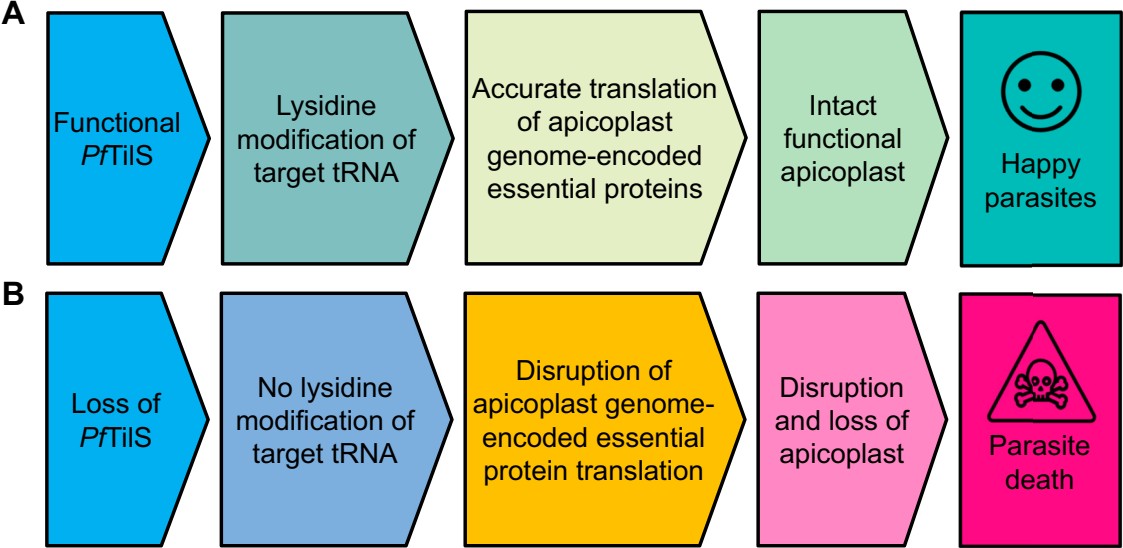

**Figure 6. Events leading to parasite death after the loss of TilS activity in the apicoplast.**

(A) TilS catalyzes lysidinylation of target tRNA_CAU to modify wobble cytidine to lysidine post-transcriptionally. This modification ensures precise decoding of isoleucine codons during apicoplast protein translation, contributing to apicoplast maintenance and parasite survival. (B) Disruption of TilS activity leads to dysfunctional protein translation within the apicoplast. This is likely a consequence of impaired isoleucine decoding caused by the absence of lysidine modification. The resulting disruption in apicoplast function ultimately culminates in parasite death.

malaria, a disease responsible for an estimated 600,000 deaths annually (World Health Organization, 2023). Recent evidence links apicoplast-localized tRNA modifications to response to frontline artemisinin drugs (Small-Saunders et al, 2024), highlighting the potential importance of tRNA modifications in parasite suscept-ibility to antimalarials. Notably, the identification of ATP-competitive TilS inhibitors in bacteria (Shapiro et al, 2014), offers a promising starting point for the discovery of novel antimalarial compounds.

## Methods

### Reagents and tools table

| Reagent/Resource | Reference or Source | Identifier or Catalog Number |
|---|---|---|
| **Experimental models** | | |
| PfMevᵃᵗᵗᴮ | Swift et al, 2021 | |

| Reagent/Resource | Reference or Source | Identifier or Catalog Number |
|---|---|---|
| **Recombinant DNA** | | |
| pCre-FH*-SFG | Rajaram et al, 2022 | |
| pCre-*pftilS-3xV5* | This study | |
| pCre-*trpftilS-3xV5* | This study | |
| pRSng | Swift et al, 2020b | |
| pCasG-LacZ | Rajaram et al, 2020 | |
| pRSng-*pftilS* | This study | |
| pCasG-*pftilSgRNA* | This study | |
| pCLD-*bsmnmA-mCherry-10xapt* | Swift et al, 2023 | |
| pCLD-*ectilS-mcherry-10xapt* | This study | |
| pINT | Nkrumah et al, 2006 | |
| **Antibodies** | | |
| Rabbit anti-mCherry | Swift et al, 2021 | |
| Mouse anti-Aldolase | David J. Sullivan, Johns Hopkins Bloomberg School of Public Health | |
| Mouse anti-V5 | Abcam | Cat # ab27671 |
| Rabbit anti-ACP | Gallagher and Prigge, 2010 | |
| Goat anti-rabbit Alexa 488 | ThermoFisher Scientific | Cat # A-11034 |
| Goat anti-mouse Alexa 594 | ThermoFisher Scientific | Cat # A-11032 |
| Donkey anti-rabbit HRP | ThermoFisher Scientific | Cat # 31458 |
| Sheep anti-mouse HRP | Millipore Sigma | Cat # GENA931 |
| **Oligonucleotides and other sequence-based reagents** | | |
| Primers | This study | Appendix Table S1 |
| Full-length *Pf*TilS | This study | Appendix Fig. S6 |
| N-terminally truncated *Pf*TilS | This study | Appendix Fig. S7 |
| **Chemicals, Enzymes, and other reagents** | | |
| Phusion High-Fidelity DNA Polymerase | ThermoFisher Scientific | Cat # F530L |
| *Avr*II | New England Biolabs Inc. | Cat # R0174S |
| *Psp*OMI | New England Biolabs Inc. | Cat # R0653S |
| *Not*I | New England Biolabs Inc. | Cat # R3189S |
| *Ngo*MIV | New England Biolabs Inc. | Cat # R0564S |
| *Bsp*EI | New England Biolabs Inc. | Cat # R0540S |
| *Bsi*WI | New England Biolabs Inc. | Cat # R3553S |
| *Bsa*I | New England Biolabs Inc. | Cat # R3733S |

| Reagent/Resource | Reference or Source | Identifier or Catalog Number |
|---|---|---|
| T4 DNA ligase | New England Biolabs Inc. | Cat # M0202 |
| In-Fusion | Clontech Laboratories | Cat # 639650 |
| RPMI 1640 medium with L-glutamine | USBiological | Cat # R8999 |
| AlbuMAX™ II Lipid-Rich BSA | ThermoFisher Scientific | Cat # 11021037 |
| Hypoxanthine | Millipore Sigma | Cat # H9377 |
| Gentamicin sulfate salt | Millipore Sigma | Cat # G3632 |
| Racemic Mevalonolactone | Millipore Sigma | Cat # M4667 |
| WR99210 | Jacobus pharmaceuticals | |
| DSM1 | BEI resources | Cat # MRA-1161 |
| Blasticidin S HCl | Corning Inc. | Cat # 30-100-RB |
| Anhydrous tetracycline (aTc) | Cayman Chemical | Cat # 10009542 |
| *Shield1* | Aobious Inc. | Cat # AOB1848 |
| ProLong Gold 4′, 6-diamidino-2-phenylindole (DAPI) antifade mountant | ThermoFisher Scientific | Cat # P36935 |
| DAPI solution | ThermoFisher Scientific | Cat # 62248 |
| MitoTracker CMX-Ros | ThermoFisher Scientific | Cat # M7512 |
| SuperSignal West Pico chemiluminescent substrate | ThermoFisher Scientific | Cat # 34577 |
| SYBR green I | ThermoFisher Scientific | Cat # S7563 |
| Saponin | Millipore Sigma | Cat # S4521 |
| NuPAGE LDS sample buffer (4x) | ThermoFisher Scientific | Cat # NP0007 |
| **Software and online resources** | | |
| Prism V8.4 | GraphPad Software | |
| Volocity | PerkinElmer | |
| ImageJ | https://imagej.net/ij/ | |
| MEGA11 | Tamura et al, 2021 | |
| UCSF ChimeraX (version: 1.5) | https://www.rbvi.ucsf.edu/chimerax | |
| UniProt Align | https://www.uniprot.org/align | |
| Boxshade | https://junli.netlify.app/apps/boxshade/ | |
| GtRNAdb | https://gtrnadb.ucsc.edu/citation.html | |
| tRNAscan-SE | https://trna.ucsc.edu/tRNAscan-SE/ | |
| UniProt | https://www.uniprot.org/ | |
| PlasmoDB | https://plasmodb.org/ | |
| ToxoDB | https://toxodb.org/ | |
| PiroplasmaDB | https://piroplasmadb.org/ | |

| Reagent/Resource | Reference or Source | Identifier or Catalog Number |
|---|---|---|
| **Other** | | |
| Gene Pulser Xcell™ electroporation system | Bio-Rad Laboratories, Inc. | |
| Zeiss AxioImager M2 microscope | Carl Zeiss Microscopy, LLC. | |
| Hamamatsu ORCA-R2 camera | Hamamatsu Photonics | |
| Attune Nxt Flow Cytometer | ThermoFisher Scientific | |

### *Plasmodium falciparum* parental parasite line

PfMev^attB parental parasites were used for generating knockout and knock-in lines. This parasite line possesses an engineered cytosolic mevalonate pathway for isoprenoid precursor production (Swift et al, 2020b; Swift et al, 2021). In addition, the apicoplast of this parasite line is labeled with a codon-optimized variant of super-folder green fluorescent protein (api-SFG) (Roberts et al, 2019).

### Asexual blood stage culture

Asexual-stage *P. falciparum* parasites were aseptically cultured in human O⁺ red blood cells (RBC) at a 2% hematocrit using RPMI 1640 medium with L-glutamine. The RPMI 1640 medium was supplemented with 12.5 μg/mL hypoxanthine, 20 mM HEPES, 0.2% sodium bicarbonate, 5 g/L Albumax II, and 25 μg/mL gentamicin. Cultures were maintained at 37 °C in 25 cm² gassed flasks with a controlled atmosphere of 94% $N_2$, 3% $O_2$, and 3% $CO_2$. Parasitemia was maintained between 2% and 5%. To ensure continuous propagation, cultures were passaged every other day by diluting with fresh medium containing uninfected RBCs.

### Construction of transfection plasmids

Codon-modified versions of the *P. falciparum tilS* (*pftilS*) gene, encoding either the full-length or N-terminally truncated protein, were synthesized by Twist Bioscience (CA, USA). Both constructs were flanked by *Avr*II and *Psp*OMI restriction enzyme sites and included a C-terminal 3xV5 epitope tag for immunodetection (sequences are available in Appendix Figs. S6, S7). The synthetic *pftilS* genes were cloned into the corresponding restriction sites within the pCre-FH*-SFG plasmid (Rajaram et al, 2022) to generate pCre-*pftilS*-3xV5 (full-length *Pf*TilS) and pCre-*trpftilS*-3xV5 (N-terminally truncated *Pf*TilS) plasmids.

The *pftilS* gene (PF3D7_0411200) was targeted for deletion using two plasmids: pRSng (Swift et al, 2020b) and pCasG-LacZ (Rajaram et al, 2020). To construct the repair plasmid (pRSng-*pftilS*), we amplified two *pftilS* homology arms (HA1, 403 bp; HA2, 569 bp) from *P. falciparum* PfMev^attB genomic DNA using specific primers (Appendix Table S1). HA1 and HA2 were inserted into the *Not*I and *Ngo*MIV restriction sites, respectively, of pRSng by In-Fusion ligation independent cloning (LIC). The guide RNA (gRNA) sequence targeting *pftilS* (20 bp) was synthesized as 5'-phosphory-lated oligonucleotides, annealed, and inserted into the *Bsa*I sites of

pCasG-LacZ using ligase dependent cloning to generate the pCasG-*pftilS*gRNA plasmid.

For generation of the *ectilS* knock-in plasmid, we amplified the *ectilS* gene (NCBI gene ID 944889) from *E. coli* genomic DNA with the following primer pair: EcTilS.InF.F and EcTilS.InF.R (Appendix Table S1). The *ectilS* amplicon was inserted into the *Bsp*EI and *Bsi*WI restriction sites via LIC, replacing the *bsmnmA* gene in plasmid pCLD-*bsmnmA*-*mcherry*-10xapt (Swift et al, 2023) to yield the pCLD-*ectilS*-*mcherry*-10xapt plasmid. Sanger and/or whole plasmid sequencing was performed on all constructs to verify sequence fidelity. All the source plasmids and the plasmids generated in this study are summarized in the Reagents and Tools table.

### Parasite transfections

To generate the Δ*pftilS* transgenic line, we transfected PfMev^attB parasites with the pRSng-*pftilS* and pCasG-*pftilS*gRNA plasmids (75 μg each) using an established transfection protocol (Spalding et al, 2010). Briefly, 400 μL of RBCs were electroporated with the plasmid mixture at low voltage using Gene Pulser Xcell™ electroporation system. The transfected RBCs were then mixed with 1.5 mL of PfMev^attB parasites (mostly mid-trophozoite to early schizont stage) and cultured in complete medium with 50 μM mevalonate (Racemic mevalonolactone) for 48 h. To select for parasites that underwent successful homologous recombination events at the *pftilS* locus, the culture medium was subsequently supplemented with 1.5 μM DSM1, 2.5 nM WR99210, and 50 μM mevalonate for a period of seven days. After this selection period, cultures were maintained in complete medium with 50 μM mevalonate until parasite emergence. Once the parasites appeared, the cultures were maintained in complete medium with 2.5 nM WR99210 and 50 μM mevalonate.

We co-transfected RBCs with pCre-*pftilS*-3xV5, pCre-*trpftilS*-3xV5, or pCLD-*ectilS*-mCherry-10xapt plasmids, respectively, with the pINT plasmid (Nkrumah et al, 2006) encoding the mycobac-teriophage integrase. The integrase facilitates recombination of the attP site found in the expression constructs with the attB site in the parasite genome. The transfected RBCs were then infected with PfMev^attB parasites (Swift et al, 2021) and cultured with 1.25 μg/mL blasticidin and 0.50 μM anhydrous tetracycline (aTc) for seven days to select for parasites with successful integration events. After seven days, cultures were maintained in complete medium with aTc until parasite emergence. Once parasites reappeared, the cultures were maintained with 1.25 μg/mL blasticidin and 0.50 μM aTc.

For *ectilS*⁺Δ*pftilS* transgenic parasite generation, we used the same Cas9 and pRSng repair plasmids used for the Δ*pftilS* line. Growth medium supplemented with 1.5 μM DSM1, 2.5 nM WR99210, 1.25 μg/mL blasticidin, and 0.50 μM aTc was used for the initial seven days of selection, after which the cultures were switched to growth medium containing blasticidin and aTc. Upon parasite appearance, all cultures were maintained in medium containing WR99210, blasticidin, and aTc.

### Genotype confirmation

Parasite lysates were prepared from parental or transgenic lines by incubation at 90 °C for 5 min. These lysates served as templates for

all subsequent genotype confirmation PCRs. For confirmation of Δ*pftilS* and *ectilS*⁺Δ*pftilS* genotypes, we used specific primer pairs (primer sequences are available in Appendix Table S1) to amplify both the 5'- and 3'-ends (designated Δ5' and Δ3', respectively) of the disrupted *pftilS* locus and the corresponding regions (designated 5' and 3', respectively) of the native locus. The expected amplicon sizes for each primer pair are provided in Fig. 3A.

For *pftilS*⁺, *trpftilS*⁺, or *ectilS*⁺ genotype confirmations, specific primers (sequences available in Appendix Table S1) were used to amplify the attL and attR recombination junctions flanking the integrated plasmids. In addition, we amplified the unaltered attB site in the parental parasite genome as a control. The anticipated sizes of the PCR products are indicated in Figs. 2B, 4A.

## Confirmation of apicoplast genome loss

We used the apicoplast-encoded *sufB* gene (PF3D7_API04700) as a proxy for detecting the apicoplast genome. The gene was amplified by PCR with a specific primer pair listed in Appendix Table S1. As controls, genes from the nuclear genome (*ldh*, PF3D7_1324900) and the mitochondrial genome (*cox1*, PF3D7_MIT02100) were amplified with corresponding primer pairs (sequences available in Appendix Table S1). Parasite lysates of the parental line were used as positive controls for apicoplast genome detection. The expected amplicon sizes for *ldh*, *sufB*, and *cox1* are 520 bp, 581 bp, and 761 bp, respectively.

## Immunoblot

Asynchronous parental and *ectilS*⁺ parasite cultures were washed three times with cold complete medium. The cultures were then treated with 0.15% (w/v) saponin in cold phosphate-buffered saline (PBS, pH 7.4) for 10 min on ice. This step permeabilizes the RBC and parasitophorous vacuolar membranes, allowing access to the intracellular parasites (Christophers and Fulton, 1939). Following saponin treatment, intact parasites were pelleted by centrifugation at $1940 \times g$ for 10 min at 4 °C. The parasite pellets were washed three additional times with cold PBS. The isolated parasites were then either used immediately or snap-frozen in liquid nitrogen and stored at −80 °C for later use.

Saponin-isolated parasites were resuspended in 1x NuPAGE LDS sample buffer containing 2% β-mercaptoethanol and boiled for 5 min to ensure complete protein denaturation and solubilization. The lysed parental and *ectilS*⁺ parasite samples were resolved by sodium dodecyl sulfate-polyacrylamide gel electrophoresis (SDS-PAGE) using 4–12% gradient reducing gels. Following SDS-PAGE, the separated proteins were transferred electrophoretically to nitrocellulose membranes. The membranes were blocked with 5% non-fat dry milk in PBS containing 0.1% Tween-20 (Milk/PBST) for 1 h at room temperature to minimize non-specific antibody binding. Following blocking, the membranes were incubated overnight at 4 °C with polyclonal rabbit anti-mCherry primary antibodies (Swift et al, 2021) (diluted 1:5000 in Milk/PBST) to detect *Ec*TilS-mCherry fusion protein (expected molecular weight 91 kDa). The membranes were then washed and incubated with polyclonal donkey anti-rabbit horseradish peroxidase (HRP)-conjugated secondary antibodies (diluted 1:10,000 in Milk/PBST) for 1 h at room temperature. Chemiluminescent signal

was developed with SuperSignal West Pico chemiluminescent substrate according to the manufacturer's instructions and detected on autoradiography film.

To ensure equal protein loading across samples, the membranes were stripped of antibodies using a 5 min incubation with 200 mM glycine buffer (pH 2.0) at room temperature. The stripped membranes were then re-blocked with 5% Milk/PBST and probed with primary anti-Aldolase mouse monoclonal antibody (diluted 1:25,000) followed by sheep anti-mouse HRP-conjugated secondary antibody (diluted 1:10,000). The chemiluminescent detection steps were repeated as described above.

## Immunofluorescence assays and live cell microscopy

For immunofluorescence assays, we fixed and permeabilized *pftilS*⁺ and *trpftilS*⁺ parasites as described previously (Gallagher et al, 2011) with minor modifications. Briefly, infected RBCs from 250 μL of culture (~5% parasitemia in 2% hematocrit) were harvested by centrifugation and resuspended in 300 μL of 4% electron microscopy (EM) grade paraformaldehyde and 0.0075% EM grade glutaraldehyde in PBS (pH 7.4) for fixation. The fixation step was carried out for 30 min at 37 °C while shaking at 225 rpm. Following fixation, the cells were permeabilized using 0.1% Triton X-100 in PBS for 10 min on a 3D-rocker. After a 2 h blocking step with 3% bovine serum albumin (BSA) in PBS at room temperature on a 3D-rocker to prevent non-specific binding, cells were incubated overnight at 4 °C with 1:500 polyclonal rabbit anti-ACP (Gallagher and Prigge, 2010) and 1:1000 monoclonal mouse anti-V5 (SV5-Pk1) primary antibodies on an orbital shaker. After the overnight incubation, the cells were washed with PBS three times and then incubated for 2 h with 1:1500 goat anti-rabbit Alexa 488 and 1:1500 goat anti-mouse Alexa 594 secondary antibodies in PBS with 3% BSA on a 3D-rocker. After three washes with PBS, the cells were mounted on coverslips with ProLong Gold 4′,6-diamidino-2-phenylindole (DAPI) antifade reagent and sealed with nail polish.

For live cell imaging of the Δ*pftilS* transgenic line, we incubated 100 μL of asynchronous parasites of ~5% parasitemia and 2% hematocrit with 1 μg/mL DAPI and 30 nM MitoTracker Red CMX-Ros for 30 min at 37 °C. Following incubation, the cells were washed three times with complete medium, with a 5 min incubation at 37 °C after each wash step. After the final wash, the parasites were resuspended in 20 μL of complete medium, placed on a slide, and sealed under a coverslip using wax. For live cell imaging of the *ectilS*⁺ and *ectilS*⁺Δ*pftilS* parasite lines, cells were stained with 1 μg/mL DAPI only.

All images were captured using a Zeiss AxioImager M2 microscope equipped with a Hamamatsu ORCA-R2 camera and a 100x/1.4 NA objective lens. A series of images were obtained spanning 5 μm along the z-axis with a spacing of 0.2 μm between each image. Subsequently, an iterative restoration algorithm implemented within Volocity software was used to deconvolve the images to report a single image in the z-plane. Red channels in all microscopy images were pseudocolored magenta using ImageJ.

## Parasite growth assay

Parasite growth was monitored using an Attune Nxt Flow Cytometer as previously described (Swift et al, 2020b; Tewari

et al, 2022). For determining the growth dependence on mevalonate presented in Fig. 3E, we cultured Δ*pftilS* parasites in the presence or absence of 50 μM mevalonate, and in the experiment presented in Fig. 5C, cultures were grown in the presence or absence of 50 μM mevalonate with 0.5 μM aTc supplementation in both conditions. In both experiments, the cultures were seeded at 0.5% initial parasitemia and 2% hematocrit in a total volume of 250 μL, in quadruplicate for each condition. Parasite growth was monitored every 24 h over four days following SYBR green I staining. For growth assays presented in Fig. 5H, we grew the *ectilS*+Δ*pftilS* parasites in the presence of 0.5 μM aTc and absence of *Shield1* (permissive condition) or in the absence of aTc and presence of 0.5 μM *Shield1* (non-permissive condition). In this experiment, parasite growth was monitored over eight days. On day four, the cultures were diluted 1:10. Data from two independent biological replicates (each in quadruplicate) of the indicated parasite lines were analyzed using a two-way ANOVA with a Sidak-Bonferroni correction in Prism V8.4.

The growth of the *ectilS*+ (Fig. 4E) and *ectilS*+Δ*pftilS* (Fig. 5F) lines were compared to that of the parental PfMev^attB parasite line. Asynchronous parasite cultures were seeded in duplicate at a standardized initial parasitemia of 0.5% and a hematocrit of 2% in 96-well plates with a total volume of 200 μL per well. In all cases, the media was supplemented with 0.5 μM aTc. Following a 48-h incubation (corresponding to a complete growth cycle), final parasitemia was determined using flow cytometry as described above. The growth rate was calculated by dividing the final parasitemia by the initial parasitemia, resulting in a fold-change (in this case fold increase) value per growth cycle. This experiment was repeated for at least five growth cycles for each parasite line and the data are presented as median fold increase per growth cycle. The two-tailed non-parametric Mann–Whitney U test was used to compare the fold increase per growth cycle between transgenic and parental lines, with an alpha level of 0.05. Statistical analysis was performed using Prism V8.4.

### Bioinformatics

Multiple sequence alignment of TilS orthologs presented in Appendix Fig. S1 was performed using the ClustalW program implemented in MEGA11 software (Tamura et al, 2021). The MSA included the following full-length protein sequences: *P. falciparum* (*Pf*TilS, PlasmoDB ID: PF3D7_0411200), *Synechocystis* sp. (*Sy*TilS, UniProt ID: P74192), *Aquifex aeolicus* (*Aa*TilS, UniProt ID: O67728), *Mycoplasma genitalium* (*Mg*TilS, UniProt ID: P47330), *E. coli* (*Ec*TilS, UniProt ID: P52097), *Geobacillus kaustophilus* (*Gk*TilS, UniProt ID: Q5L3T3), and *Arabidopsis thaliana* (*At*RSY3, UniProt ID: F4J7P7). Sequence alignment between *Pf*TilS and *Archaeoglobus fulgidus* TiaS (*Af*TiaS, UniProt ID: O28025) (Appendix Fig. S5) was done using the Align tool in UniProt. Alignment outputs were visualized with Boxshade to highlight conserved residues.

Crystal structure of *E. coli* TilS and AlphaFold predicted *P. falciparum* TilS structure in Fig. 1F were visualized with UCSF ChimeraX (Pettersen et al, 2021).

Phylogenetic analyses presented in Appendix Fig. S3 and Fig. EV2 were performed using MEGA11 software (Tamura et al, 2021). We used the maximum likelihood method incorporating the JTT matrix-based model (Jones et al, 1992). To assess the robustness of the inferred tree topology, 1000 bootstrap replicates were performed. The resulting tree with the highest likelihood score is presented. The proteins used for phylogenetic analysis presented in Appendix Fig. S3 are listed in Appendix Table S2, and the tRNAs used for analysis presented in Fig. EV2 are listed in Appendix Table S3 with accession ID where available. Sequences of tRNA genes with anticodons for methionine (CAU) and isoleucine (CAU, AAU, UAU, GAU) were obtained from GtRNAdb (Chan and Lowe, 2015) for *H. marismortui*, *Mycoplasma mobile*, *Bacillus subtilis*, *E. coli*, *Cycas panzhihuaensis*, *Solanum lycopersicum*, and *Saccharomyces cerevisiae*. Apicoplast tRNA sequences for *P. falciparum* were retrieved from PlasmoDB (Aurrecoechea et al, 2008), *Toxoplasma gondii* and *Eimeria tenella* from ToxoDB (Kissinger et al, 2003), and *Babesia microti* and *Theileria parva* from PiroplasmaDB (Alvarez-Jarreta et al, 2023).

The secondary structures of tRNAs presented in Fig. EV3 were generated using tRNAscan-SE online program (Lowe and Chan, 2016).

### Biosafety

All *P. falciparum* experiments were performed in a Class II biosafety cabinet equipped with a HEPA filter, in accordance with institutional safety guidelines and approved by the Johns Hopkins University Office of Health, Safety, and Environment.

## Data availability

The raw microscopy image data are available in the BioImage Archive under accession number S-BIAD1577.

The source data of this paper are collected in the following database record: biostudies:S-SCDT-10_1038-S44319-025-00420-w.

## Peer review information

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

## Acknowledgements

We are grateful to David J. Sullivan (Johns Hopkins Bloomberg School of Public Health) for his generous gift of the mouse anti-aldolase monoclonal antibody.

We extend our sincere appreciation to the students enrolled in Biology of Parasitism 2021 for their contributions to this study. Fluorescent imaging was conducted with the support and instruments of the Light Microscopy Core within the Department of Molecular Microbiology and Immunology. In addition, we extend our appreciation to the Johns Hopkins Malaria Research Institute Parasite Core for supplying O⁺ human red blood cells for parasite culture. Financial support for this research was provided by the National Institutes of Health grant R01 AI125534 (STP), the Johns Hopkins Malaria Research Institute postdoctoral fellowship (RE), Samuel Jordan Graham postdoctoral fellowship (RE), the Johns Hopkins Malaria Research Institute, and Bloomberg Philanthropies. The funding agencies did not exert any influence on the design of the study, the collection or analysis of data, the decision to publish the findings, or the preparation of the manuscript.

## Author contributions

**Rubayet Elahi**: Conceptualization; Data curation; Formal analysis; Investigation; Visualization; Methodology; Writing—original draft; Writing—review and editing. **Sean T Prigge**: Conceptualization; Formal analysis; Supervision; Funding acquisition; Project administration; Writing—review and editing.

Source data underlying figure panels in this paper may have individual authorship assigned. Where available, figure panel/source data authorship is listed in the following database record: biostudies:S-SCDT-10_1038-S44319-025-00420-w.

## Disclosure and competing interests statement

The authors declare no competing interests.

# Expanded View Figures

| aa | Codon | Frequency/ 1000 (Number) | aa | Codon | Frequency/ 1000 (Number) | aa | Codon | Frequency/ 1000 (Number) | aa | Codon | Frequency/ 1000 (Number) |
|---|---|---|---|---|---|---|---|---|---|---|---|
| F | UUU | 61.7 (458) | | UCU | 17.7 (131) | Y | UAU | 104.8 (778) | C | UGU | 10.9 (81) |
| | UUC | 0.7 (5) | S | UCC | 0.7 (5) | | UAC | 1.1 (8) | | UGC | 0.1 (1) |
| L | UUA | 119.6 (888) | | UCA | 13.1 (97) | | UAA | 3.1 (23) | | UGA | 0.9 (7) |
| | UUG | 1.6 (19) | | UCG | 0.4 (3) | | UAG | - | W | UGG | 3.9 (29) |
| L | CUU | 0.5 (4) | P | CCU | 10.2 (76) | H | CAU | 9.4 (70) | R | CGU | 1.9 (14) |
| | CUC | - | | CCC | 0.1 (1) | | CAC | - | | CGC | - |
| | CUA | 1.4 (10) | | CCA | 3.6 (27) | Q | CAA | 15.8 (117) | | CGA | - |
| | CUG | - | | CCG | 0.1 (1) | | CAG | 1.2 (9) | | CGG | - |
| I | AUU | 70.1 (520) | T | ACU | 14.3 (106) | N | AAU | 150.5 (1117) | S | AGU | 14.6 (108) |
| | AUC | 1.1 (8) | | ACC | 0.1 (1) | | AAC | 0.9 (7) | | AGC | - |
| | AUA | 102.3 (759) | | ACA | 13.6 (101) | K | AAA | 123.7 (918) | R | AGA | 11.3 (84) |
| M | AUG | 11.6 (86) | | ACG | 0.3 (2) | | AAG | 7.1 (53) | | AGG | 0.4 (3) |
| V | GUU | 6.6 (49) | A | GCU | 5.5 (41) | D | GAU | 17.3 (128) | G | GGU | 16.0 (119) |
| | GUC | - | | GCC | 0.1 (1) | | GAC | 0.3 (2) | | GGC | - |
| | GUA | 10.6 (79) | | GCA | 2.2 (16) | E | GAA | 19.5 (145) | | GGA | 11.5 (85) |
| | GUG | 0.1 (1) | | GCG | - | | GAG | 1.1 (8) | | GGG | 1.8 (13) |

**Figure EV1.  Codon usage by 30 proteins encoded in the *P. falciparum* apicoplast genome.**

The table presents codon usage frequencies for all proteins encoded by the *P. falciparum* apicoplast genome. Frequencies are expressed as the number of occurrences of each codon per thousand codons, with the total number of occurrences provided in parentheses. Stop codons are shaded in gray. AUA (highlighted in yellow) is the most frequently used codon for isoleucine (I) in apicoplast-genome-encoded proteins. Single-letter amino acid (aa) codes are used. A dash (–) indicates codons that are not used in apicoplast-genome-encoded proteins.

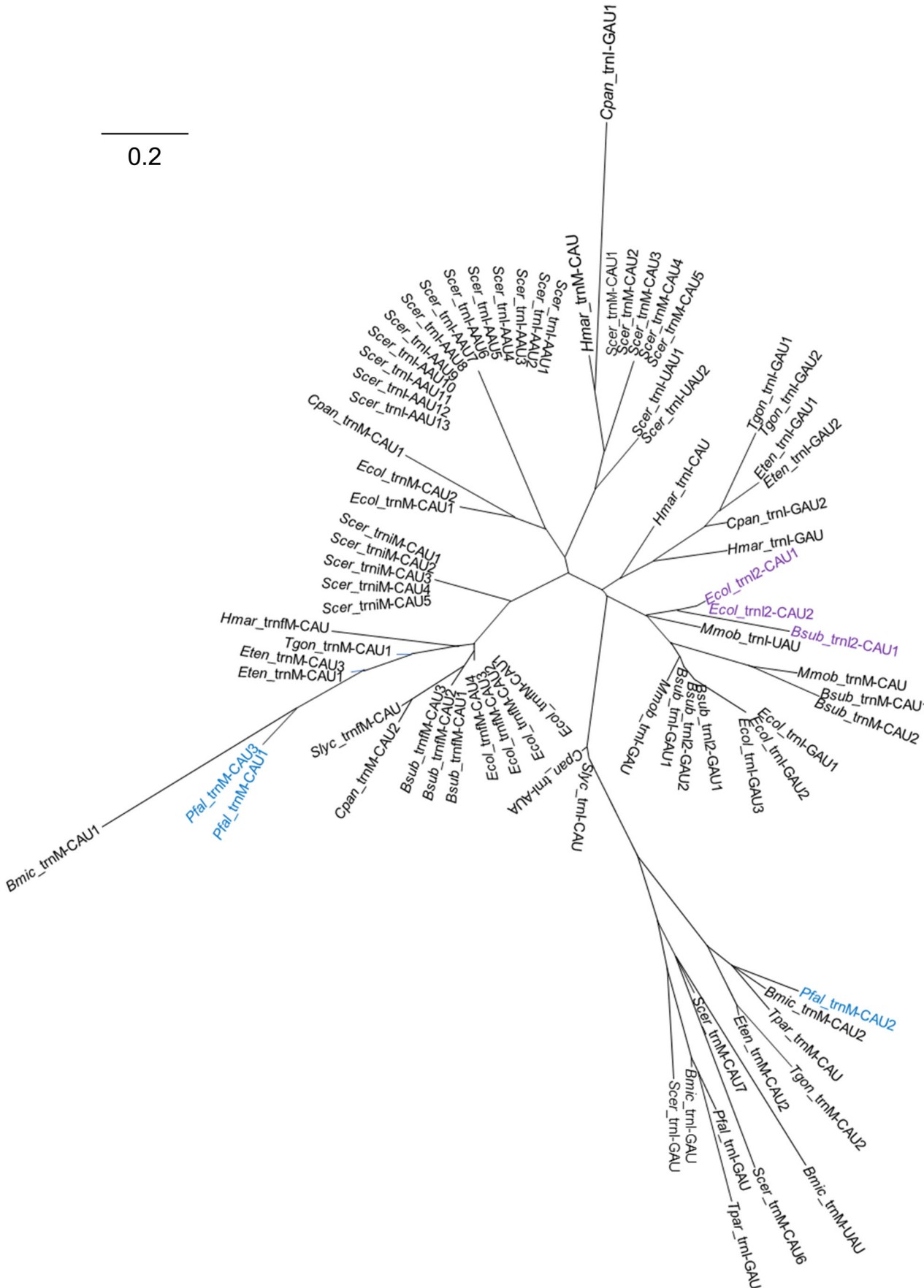

◄ **Figure EV2. The phylogenetic relationship among methionine- and isoleucine-decoding tRNAs.**

Apicoplast encoded tRNA$_{CAU}$ are shown in blue font with PF3D7_API00600 (*pfal*_trnM-CAU2) sharing common ancestry with experimentally validated TilS substrate tRNA$^{Ile}_{CAU}$ from other species (in purple font). Bootstrap analyses (1000 replicates) were employed to assess the robustness of the branching patterns. The phylogenetic tree is drawn to scale, with branch lengths representing the estimated number of substitutions per site. Refer to Appendix Table S3 for tRNA sequences.

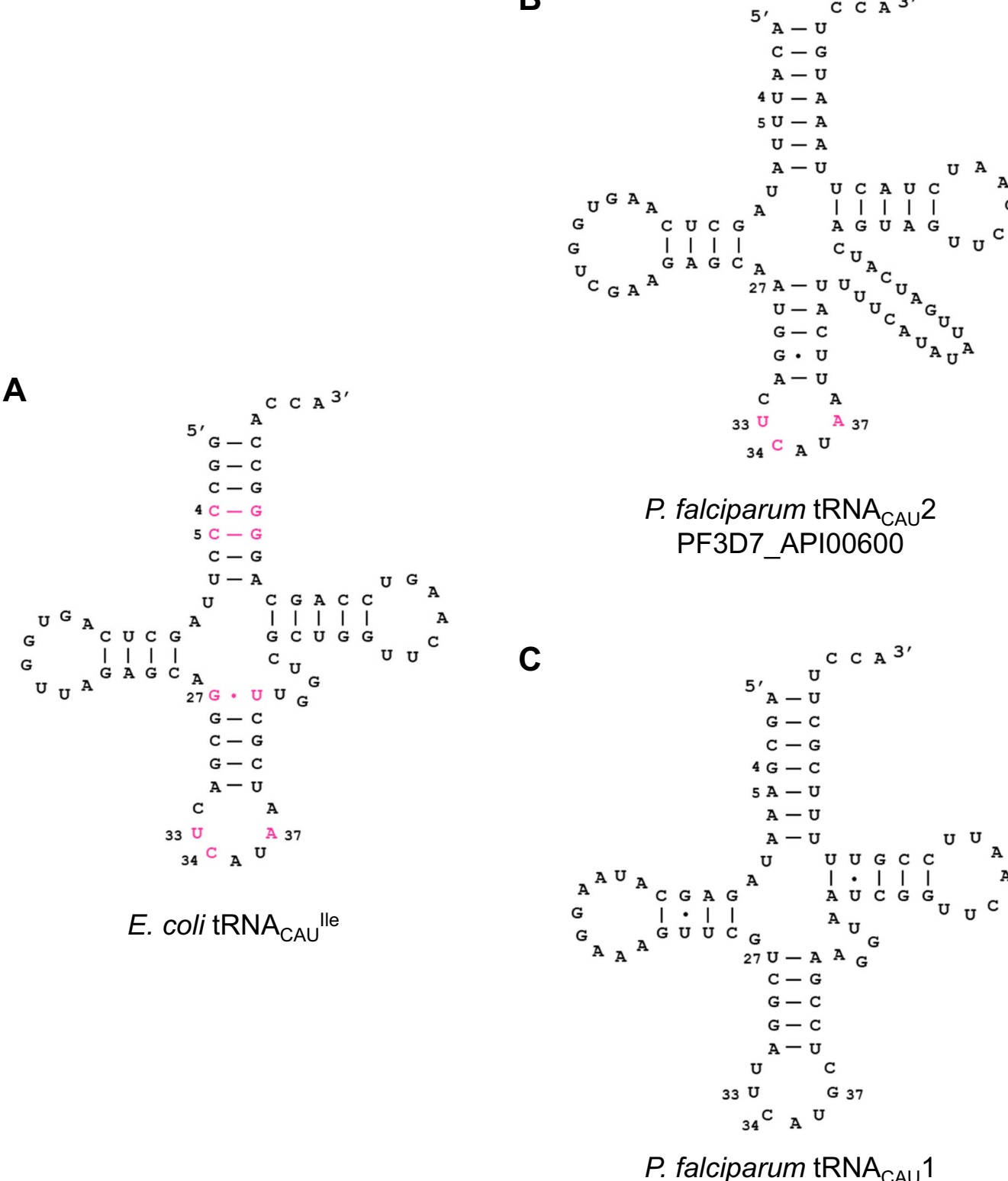

**A**

*E. coli* tRNA_CAU^Ile

**B**

*P. falciparum* tRNA_CAU2
PF3D7_API00600

**C**

*P. falciparum* tRNA_CAU1
PF3D7_API06600

*P. falciparum* tRNA_CAU3
PF3D7_API05000

◀  **Figure EV3. Comparison of secondary structures between *Escherichia coli* tRNA^Ile_CAU and *Plasmodium falciparum* tRNAs from the apicoplast genome which are currently annotated as tRNA^Met_CAU.**

(A) *E. coli* tRNA^Ile_CAU. (B) PF3D7_API00600. (C) PF3D7_API06600 and PF3D7_API05000 (identical sequences due to gene duplication). In (A) and (B), the positive determinants for TilS binding are shown in magenta. In all panels, the dash (–) indicates canonical base pairing and the dot (.) indicates GU or UU base pairing.

