## [Peer Review File · EMBO Reports]

tRNA lysidinylation is essential for the minimal translation system in the *Plasmodium falciparum* apicoplast

Rubayet Elahi and Sean Prigge

Corresponding author(s): Sean Prigge (sprigge2@jhu.edu), Rubayet Elahi (aelahi3@jhu.edu)

Review Timeline:

Submission Date:	18th Oct 24
Editorial Decision:	4th Dec 24
Revision Received:	21st Jan 25
Editorial Decision:	3rd Feb 25
Revision Received:	27th Feb 25
Accepted:	5th Mar 25

Editor: Achim Breiling

Transaction Report:

Dear Dr. Prigge,

Thank you for the transfer of your manuscript to EMBO reports. I have now received the reports from the three referees that were asked to evaluate your study, which can be found at the end of this email.

As you will see, the referees think that these findings are of interest. They have some comments, concerns, and suggestions, indicating that a revision of the manuscript is necessary to allow publication of the study in EMBO reports. As the reports are below, and all the referee concerns need to be addressed, I will not detail them here.

However, I strongly feel that the main claim of let-7-mediated regulation of AGO1 through a classical post-transcriptional silencing pathway needs to be consolidated, e.g. by investigating changes in both AGO1 mRNA and protein stability in the context of your mutants, or by validating AGO2 association with AGO1's 3'UTR using RIP experiments, thus ruling out the possibility of a translational read-through mechanisms.

Given the constructive referee comments, I would like to invite you to revise your manuscript with the understanding that the concerns of the referees must be addressed in the revised manuscript and in a detailed point-by-point response. Acceptance of your manuscript will depend on a positive outcome of a second round of review. It is EMBO reports policy to allow a single round of revision only and acceptance of the manuscript will therefore depend on the completeness of your responses included in the next, final version of the manuscript.

- 1) a .docx formatted version of the final manuscript text (including legends for main figures, EV figures and tables), but without the figures included. Figure legends should be compiled at the end of the manuscript text.
- 2) individual production quality figure files as .eps, .tif, .jpg (one file per figure), of main figures and EV figures. Please upload these as separate, individual files upon re-submission.

4) a complete author checklist, which you can download from our author guidelines (<https://www.embopress.org/page/journal/14693178/authorguide>). Please insert page numbers in the checklist to indicate where the requested information can be found in the manuscript. The completed author checklist will also be part of the RPF.

Please also follow our guidelines for the use of living organisms, and the respective reporting guidelines: <http://www.embopress.org/page/journal/14693178/authorguide#livingorganisms>

5) that primary datasets produced in this study (e.g. RNA-seq, ChIP-seq, structural and array data) are deposited in an appropriate public database. If no primary datasets have been deposited, please also state this in a dedicated section (e.g. 'No primary datasets have been generated and deposited'), see below.

The accession numbers and database should be listed in a formal "Data Availability" section that follows the model below. This is now mandatory (like the COI statement). Please note that the Data Availability Section is restricted to new primary data that are part of this study. This section is mandatory. As indicated above, if no primary datasets have been deposited, please state this in this section

Data availability

8) Regarding data quantification and statistics, please make sure that the number "n" for how many independent experiments were performed, their nature (biological versus technical replicates), the bars and error bars (e.g. SEM, SD) and the test used to calculate p-values is indicated in the respective figure legends (also for EV and Appendix figures). Please also check that all the p-values are explained in the legend, and that these fit to those shown in the figure. Please provide statistical testing where applicable. Please avoid the phrase 'independent experiment', but clearly state if these were biological or technical replicates. Please also indicate (e.g. with n.s.) if testing was performed, but the differences are not significant. In case n=2, please show the data as separate datapoints without error bars and statistics. See also: <http://www.embopress.org/page/journal/14693178/authorguide#statisticalanalysis>

If $n < 5$, please show single datapoints for diagrams.

9) Please add scale bars of similar style and thickness to microscopic images, using clearly visible black or white bars (depending on the background). Please place these in the lower right corner of the images themselves. Please do not write on or near the bars in the image but define the size in the respective figure legend.

10) Please also note our reference format:

12) We now use CRediT to specify the contributions of each author in the journal submission system. CRediT replaces the author contribution section. Please use the free text box to provide more detailed descriptions and do NOT provide your final manuscript text file with an author contributions section. See also our guide to authors:

<https://www.embopress.org/page/journal/14693178/authorguide#authorshipguidelines>

13) All Materials and Methods need to be described in the main text using our 'Structured Methods' format, which is required for all research articles. According to this format, the Methods section should include a Reagents and Tools Table (listing key reagents, experimental models, software, and relevant equipment and including their sources and relevant identifiers), uploaded as separate file, and a Methods section in which we encourage the authors to describe their methods using a step-by-step protocol format with bullet points, to facilitate the adoption of the methodologies across labs. More information on how to adhere to this format as well as downloadable templates (.doc) for the Reagents and Tools Table can be found in our author guidelines (section 'Structured Methods'):

14) Please order the manuscript sections like this, using these names:

Title page - Abstract - Keywords - Introduction - Results - Discussion - Methods - Data availability section - Acknowledgements (including funding information) - Disclosure and Competing Interests Statement - References - Figure legends - Expanded View Figure legends

15) Please make sure that all the funding information is also entered into the online submission system and that it is complete and similar to the one in the acknowledgement section of the manuscript text file.

I look forward to seeing a revised form of your manuscript when it is ready.

Yours sincerely,

Referee #1:

The authors found in this work a *Plasmodium falciparum* gene encoding a protein that is similar to tRNA lysidine synthase (TlIS), which distributes widely among eubacteria catalyzing the anticodon modification of AUA-specific isoleucine tRNAs. They

successfully proved that this PfTiIS protein is localized in apicoplast, a plastid-like organelle, and that a loss of its function is complemented by a bacterial TiIS imported into the organelle. This is the first demonstration that the TiIS function is present in eukaryotes, though some tiis-like genes had been identified in other eukaryotes. As the TiIS function is essential for the maintenance of apicoplast in the parasite (and as human does not have this function), it can be a hopeful target of the therapy against malaria. Therefore, this work surely has a great impact on the communities of molecular and synthetic biologists and probably also on the researchers in the field of malaria therapy. On the other hand, the authors mention nothing about another enzyme TiaS that catalyze agmatidine synthesis on AUA-specific isoleucine tRNA species in archaea. It is clear from the sequence alignments that PfTiIS is distinct from TiaS, but the wet experiments including the complementation test do not by themselves exclude the possibility that PfTiIS might catalyze a similar but different reaction. I do not think this is a big defect of the manuscript, considering that the authors adequately explain about the difficulty associated with the identification of the trace amount of lysidine (and other modified nucleosides) in apicoplast tRNA. The presumed origin of apicoplasts from plastids of some algae also suggest that the enzyme is different from TiaS.

1. What are the major claims and how significant are they?

The authors claims that they have found a plasmmodium gene that functions in apicoplast as tRNA lysidine synthetase. This is the first identification of the TiIS function in eukaryotes.

2. Are the claims novel and convincing?

Yes.

3. Are the claims appropriately discussed in the context of earlier literature?

Yes, except that authors may have misidentified the paper in which lysidine had first been characterized.

4. Who will be interested and why?

Researchers in the field of genetic code evolution, tRNA, tRNA modification, aminoacyl-tRNA synthetases, and mRNA translation in general will be interested because most of them had no idea about lysidine in eukaryote. Malaria researchers may also be interested because human does not have the TiIS function, which suggests that PfTiIS could be a hopeful therapeutic target.

5. Does the paper stand out in some way from the others in its field?

Yes.

6. Are the experimental data of sufficient quality to justify the conclusions?

Probably yes. The methods are very specific to the research of malaria parasites, and I cannot tell the quality standard in this field. But the data are clear.

Minor points:

1. Page 4, line 27:

Lysidine was first characterized in Ref. 22 (Muramatsu et al., 1989, JBC). Thus, this JBC paper should be cited in this context.

2. The first paragraph of Discussion:

Another enzyme tRNA agmatidine synthase (TiaS) is known to catalyze the modification on the CAU anticodon of isoleucine tRNAs in archaea. Agmatidine is chemically similar to lysidine, while the TiaS enzyme is clearly distinct from TiIS. The function of agmatidine modification on the CAU anticodon is indistinguishable from that of lysidine modification. Therefore, the passage "TiIS, the only known enzyme catalyzing lysidine modification at the anticodon wobble position of tRNA^{lle}CAU." is misleading, though it is correct. I could not find any mention about TiaS throughout the text. Archaeal tRNA^{lle}CAU genes had been misannotated in earlier times as mentioned by the authors in Page 12, probably because they were not the substrate of TiIS, but that of TiaS.

Referee #2:

This manuscript by Elahi and Prigge characterises the tRNA isoleucine lysidine synthetase (TiIS) ortholog (PfTiIS) of the apicoplast (an organelle of prokaryotic origin) of Plasmodium falciparum parasites.

TiIS is responsible for lysidine modification of the cytidine at wobble position (C34) of tRNA(CAU) directing it to pair with AUA and decode to isoleucine rather than methionine (AUG), to ensure accurate protein synthesis. It is not present in humans and represents a potential therapeutic target.

The authors showed that tagged PfTiIS is trafficked to the apicoplast in transfectant parasites. CRISPR/Cas9 mediated gene deletion led to apicoplast disruption and loss of the organellar genome. The authors also showed that Escherichia coli TiIS can functionally complement the loss of PfTiIS.

The study is carefully performed and provides a useful analysis of the role of PfTiIS.

The study confirms that this prokaryotic origin gene in the apicoplast is essential, with its absence leading to slow death phenotype. This is consistent with previous studies that analysed other apicoplast directed enzymes involved in protein translation.

The authors claim that the work informs synthetic biology efforts to design highly reduced biological systems; however, the study has not been able to unambiguously identify the tRNA substrate for PfTilS, and does not provide an insights into why AUA is the most frequently used codon for Ile for the proteins encoded in the apicoplast genome.

Thus the work is useful but limited in scope.

Referee #3:

This is a well conducted, discrete study that shows that the Plasmodium apicoplast depends on an apicoplast-localised PfTilS homologue. Sound CRISPR experiments confirm that the gene is essential for parasite growth, and the well-designed mevalonate rescue experiments confirm that the growth defect is due to an apicoplast function. A robust complementation experiment with an E. coli TilS confirms that the function lost in the PfTilS mutant was tRNA lysidinylation, as predicted from the bioinformatic analysis of the Pf gene. This is a convincing study that will be of interest to scientists interested in tRNA biology, organellar genomes or Plasmodium cell biology.

- The Domain analysis is sound, although unclear how good the match is to N-terminus of bacterial TilS homologues.
- The apicoplast localisation of the PfTilS looks convincing. The leader sequence truncation adds to the confidence of the localisation.
- The knockout is well done and shows essentiality, and the mevalonate dependence in the authors cleverly-made lines shows that this essentiality is linked to an apicoplast function
- The EcTilS complementation is a nice experiment - I can't think of cases of this having been performed in Plasmodium before, at least not for the apicoplast - a neat approach.

Some minor issues for addressing:

- Text clarification in the abstract:

"We identified a TilS ortholog (PfTilS) *located* in the apicoplast of Plasmodium falciparum parasites"

- This located could confuse some readers into a misunderstanding that the orthologous gene is located in the apicoplast (on the apicoplast genome), where the authors intend to convey the meaning that the protein is targeted to the compartment. Reword for clarity (eg "We identified a TilS ortholog (PfTilS) *targeted to* the apicoplast...?_"

- Missing Citation

I think the relevant citation for the apicoplast genome (which reveals the total number of tRNAs is Wilson et al 1996 J Mol Biol, Complete gene map of the plastid-like DNA of the malaria parasite Plasmodium falciparum

- Please define "superwobbling"

- The relevant uniprot ids are provided for the MSA, but not for the genes that make up the phylogenetic trees in supp figure 3 or 5. The IDs for all sequences should be provided.

- Supplementary figure 2 uses symbols and highlights that are not defined in the figure legend. Please expand.

We thank the reviewers for their overall enthusiasm and helpful critique of our manuscript. In response to the reviewers' comments, we conducted additional analyses and added two additional figures (Fig. EV1 and Appendix Fig. S5) to the manuscript. Appropriate text changes (highlighted by red text in the "Related Manuscript File") were also made in response to comments from all three reviewers.

Reviewer 1 Critique:

The authors found in this work a Plasmodium falciparum gene encoding a protein that is similar to tRNA lysidine synthase (TilS), which distributes widely among eubacteria catalyzing the anticodon modification of AUA-specific isoleucine tRNAs. They successfully proved that this PfTilS protein is localized in apicoplast, a plastid-like organelle, and that a loss of its function is complemented by a bacterial TilS imported into the organelle. This is the first demonstration that the TilS function is present in eukaryotes, though some tilS-like genes had been identified in other eukaryotes. As the TilS function is essential for the maintenance of apicoplast in the parasite (and as human does not have this function), it can be a hopeful target of the therapy against malaria. Therefore, this work surely has a great impact on the communities of molecular and synthetic biologists and probably also on the researchers in the field of malaria therapy. On the other hand, the authors mention nothing about another enzyme TiaS that catalyze agmatidine synthesis on AUA-specific isoleucine tRNA species in archaea. It is clear from the sequence alignments that PfTilS is distinct from TiaS, but the wet experiments including the complementation test do not by themselves exclude the possibility that PfTilS might catalyze a similar but different reaction. I do not think this is a big defect of the manuscript, considering that the authors adequately explain about the difficulty associated with the identification of the trace amount of lysidine (and other modified nucleosides) in apicoplast tRNA. The presumed origin of apicoplasts from plastids of some algae also suggest that the enzyme is different from TiaS.

We thank the reviewer for their thoughtful and encouraging feedback on our study and for the suggestion to add information about TiaS to our manuscript. In archaea, TiaS catalyzes the modification of cytidine (C34) to agmatinylycytidine (agm²C, agmatidine) in tRNA_{CAU}, enabling accurate decoding of the AUA codon as isoleucine while avoiding misreading of AUG. While agmatidine and lysidine are chemically similar, the catalytic domains and mechanisms of TiaS and TilS are significantly distinct. To address the reviewer's comment, we performed a sequence alignment of PfTilS with a TiaS protein and found little conservation in residues critical for agmatinylation activity. This sequence alignment has now been included as a supplementary figure (Appendix Figure S5).

Together with our domain analysis of PfTilS and functional complementation with EcTilS, this sequence analysis strongly supports the conclusion that PfTilS functions as a lysidine synthase. Now, we have added a paragraph (lines 344–357) explaining how *P. falciparum* TilS differs from archaeal TiaS, supporting its role in lysidinylation rather than agmatinylation of tRNA.

Minor points:

1. Page 4, line 27:

Lysidine was first characterized in Ref. 22 (Muramatsu et al., 1989, JBC). Thus, this JBC paper should be cited in this context.

We added the correct citation as suggested by the reviewer.

2. The first paragraph of Discussion:

Another enzyme tRNA agmatidine synthase (TiaS) is known to catalyze the modification on the CAU anticodon of isoleucine tRNAs in archaea. Agmatidine is chemically similar to lysidine, while the TiaS enzyme is clearly distinct from TiiS. The function of agmatidine modification on the CAU anticodon is indistinguishable from that of lysidine modification. Therefore, the passage "TiiS, the only known enzyme catalyzing lysidine modification at the anticodon wobble position of tRNA^{Ile}CAU." is misleading, though it is correct. I could not find any mention about TiaS throughout the text. Archaeal tRNA^{Ile}CAU genes had been misannotated in earlier times as mentioned by the authors in Page 12, probably because they were not the substrate of TiiS, but that of TiaS.

We thank the reviewer for pointing out the need to clarify the distinction between TiiS and TiaS and their respective modifications. We agree that agmatidine and lysidine modifications serve functionally similar roles in ensuring accurate decoding of the AUA codon as isoleucine, while preventing misreading of AUG. However, we respectfully disagree with the notion that the statement "TiiS, the only known enzyme catalyzing lysidine modification at the anticodon wobble position of tRNA^{Ile}CAU" is misleading. This statement accurately describes the specific catalytic activity of TiiS, which is distinct from the analogous activity of archaeal TiaS.

We acknowledge that the absence of a discussion on TiaS in the original manuscript may have caused confusion. To address this, we have now added a dedicated section in the discussion (lines 344–357) explaining the differences between *P. falciparum* TiiS and archaeal TiaS, supporting the role of TiiS in lysidinylation rather than agmatinylation of tRNA. To further substantiate this, we performed a sequence alignment between *P. falciparum* TiiS and *Archaeoglobus fulgidus* TiaS, which shows poor conservation of catalytically important residues. This alignment has been included as Appendix Figure S5. We hope these additions address the reviewer's concern and provide clarity on the distinct evolutionary and biochemical roles of TiiS and TiaS.

Reviewer 2 Critique:

This manuscript by Elahi and Prigge characterises the tRNA isoleucine lysidine synthetase (TiiS) ortholog (PFTiiS) of the apicoplast (an organelle of prokaryotic origin) of Plasmodium falciparum parasites.

TiiS is responsible for lysidine modification of the cytidine at wobble position (C34) of tRNA(CAU) directing it to pair with AUA and decode to isoleucine rather than methionine (AUG), to ensure accurate protein synthesis. It is not present in humans and represents a potential therapeutic target.

The authors showed that tagged PFTiiS is trafficked to the apicoplast in transfectant parasites. CRISPR/Cas9 mediated gene deletion led to apicoplast disruption and loss of the organellar genome. The authors also showed that Escherichia coli TiiS can functionally complement the loss of PFTiiS.

The study is carefully performed and provides a useful analysis of the role of PFTiiS.

The study confirms that this prokaryotic origin gene in the apicoplast is essential, with its absence leading to slow death phenotype. This is consistent with previous studies that analysed other apicoplast directed enzymes involved in protein translation.

The authors claim that the work informs synthetic biology efforts to design highly reduced biological systems; however, the study has not been able to unambiguously identify the tRNA substrate for PFTiiS, and does not provide an insights into why AUA is the most frequently used codon for Ile for the proteins encoded in the apicoplast genome.

Thus the work is useful but limited in scope.

We acknowledge the importance of experimentally validating the tRNA substrate for *Pf*TiIS. While it is true that experimental validation was not performed in this study, we have made substantial efforts to provide insight into which of the three CAU anticodon-bearing apicoplast tRNAs could potentially serve as the substrate for *Pf*TiIS through bioinformatics analysis (Figures EV2 and EV3). Our approach mirrors a previously established method used to identify the tRNA substrate for TiaS, the analogous enzyme in archaea that catalyzes a chemically similar modification to lysidine (agmatidine) at C34 of tRNA^{Ile}_{CAU} (PMID: 17998287). This method accurately predicted the tRNA substrate for TiaS, which was subsequently validated experimentally in the same study. Given the success of this approach, we believe that our bioinformatic prediction provides a solid base for identifying the putative tRNA substrate for *Pf*TiIS.

We appreciate the reviewer's query on why AUA is the predominant codon for isoleucine in apicoplast-genome-coded proteins. As we discussed in this revised version of the manuscript (lines 408-419), this preference can be attributed to the high AT-rich nucleotide composition (89.1%) of the protein-coding genes. In such high AT-rich environments, A or T-ending codons are favored, a pattern that is well-documented in other AT-rich endosymbiont genomes. Additionally, many endosymbionts exhibit a similar preference for AUA to decode isoleucine, likely reflecting an evolutionary adaptation of TiIS to efficiently recognize AUA in these AT-rich contexts. To help highlight the points made about the apicoplast genome, we added Figure EV1 with a complete apicoplast codon table.

Reviewer 3 Critique:

This is a well conducted, discrete study that shows that the Plasmodium apicoplast depends on an apicoplast-localised PfTiIS homologue. Sound CRISPR experiments confirm that the gene is essential for parasite growth, and the well-designed mevalonate rescue experiments confirm that the growth defect is due to an apicoplast function. A robust complementation experiment with an E. coli TiIS confirms that the function lost in the PfTiIS mutant was tRNA lysidinylation, as predicted from the bioinformatic analysis of the Pf gene. This is a convincing study that will be of interest to scientists interested in tRNA biology, organellar genomes or Plasmodium cell biology.

- The Domain analysis is sound, although unclear how good the match is to N-terminus of bacterial TiIS homologues.*
- The apicoplast localisation of the PfTiIS looks convincing. The leader sequence truncation adds to the confidence of the localisation.*
- The knockout is well done and shows essentiality, and the mevalonate dependence in the authors cleverly-made lines shows that this essentiality is linked to an apicoplast function.*
- The EcTiIS complementation is a nice experiment - I can't think of cases of this having been performed in Plasmodium before, at least not for the apicoplast - a neat approach.*

We agree that the long insertions typically found in *P. falciparum* proteins make it challenging to analyze domain conservation in pairwise alignments, and TiIS is no exception. However, our multiple sequence alignment of TiIS orthologs, presented in Supplementary Figure 2 in the original manuscript (now Appendix Figure S1), highlights the conservation of key residues critical for lysidinylation activity and provides a good indication of where low complexity sequences are inserted. Additionally, as presented in Figure 1F, structural predictions using AlphaFold suggest that there could be a similar folding pattern for the individual domains, indicating how the low complexity regions could form loops which do not interfere with the domain structure.

Some minor issues for addressing:

- Text clarification in the abstract:

*"We identified a TiIS ortholog (PFTiIS) *located* in the apicoplast of Plasmodium falciparum parasites" This located could confuse some readers into a misunderstanding that the orthologous gene is located in the apicoplast (on the apicoplast genome), where the authors intend to convey the meaning that the protein is targeted to the compartment. Reword for clarity (eg "We identified a TiIS ortholog (PFTiIS) *targeted to* the apicoplast...?_"*

The corrections were made as suggested. Now lines 35-36 in the abstract reads as follows: "We identified a TiIS ortholog (PFTiIS) targeted to the apicoplast of *Plasmodium falciparum* parasites."

- Missing Citation

I think the relevant citation for the apicoplast genome (which reveals the total number of tRNAs is Wilson et al 1996 J Mol Biol, Complete gene map of the plastid-like DNA of the malaria parasite Plasmodium falciparum

Corrected.

- Please define "superwobbling"

We added the definition of superwobbling at its first mention in line 95. The revised text in lines 94–96 now reads: "...maximal use of wobble and superwobbling (where a tRNA with an unmodified U at the wobble position recognizes all four nucleotides at the third codon position) base pairing.....".

- The relevant uniprot ids are provided for the MSA, but not for the genes that make up the phylogenetic trees in supp figure 3 or 5. The IDs for all sequences should be provided.

The UniProt entries for the proteins included in the phylogenetic tree in Supplementary Figure 3 (now Appendix Figure S3) are listed in Supplementary Table 2 (now Appendix Table S2). Similarly, the tRNA sequences used to construct the phylogenetic tree presented in Supplementary Figure 5 (now Figure EV2) are detailed in Supplementary Table 3 (now Appendix Table S3), including accession IDs where available. However, the majority of sequences obtained from the tRNA database (GtRNAdb) do not have unique accession IDs. To address this, we updated the methods section (lines 660–671) to provide a comprehensive description of the sources for all sequences used in both phylogenetic analyses.

- Supplementary figure 2 uses symbols and highlights that are not defined in the figure legend. Please expand.

We added the figure legends directly within the figure (now Appendix Figure S1) for clarity.

Dear Dr. Prigge,

Thank you for the submission of your revised manuscript to our editorial offices. I have now received the report from the three referees that were asked to re-evaluate the study, you will find below. As you will see, the referees now fully support the publication of the study in EMBO reports.

Before I can proceed with formal acceptance, I have these editorial requests I ask you to address in a final revised manuscript:

- Please provide a comprehensive final title with not more than 100 characters (including spaces).
- Please provide the abstract written in present tense throughout.
- We plan to publish your manuscript as Report. For a Scientific Report we require that results and discussion sections are combined in a single chapter called "Results & Discussion". Please do this for your manuscript. Moreover, please rearrange the main figures so that there will be 5 final main figures (plus up to 5 RV figures). Please then carefully update all the figure callouts and make sure that all panels are called out and that they are called out sequentially. Presently, there seem to be no separate callouts for panels 6A and 6B.

For more details please refer to our guide to authors:

<http://www.embopress.org/page/journal/14693178/authorguide#researcharticleguide>

- Please remove the Synopsis and the bullet points from the manuscript text. I have saved these separately and forward these to our publisher after acceptance of the manuscript.

- Please make check again that the number "n" for how many independent experiments were performed, their nature (biological versus technical replicates), the bars and error bars (e.g. SEM, SD) and the test used to calculate p-values is indicated in the respective figure legends. Please also check that all the p-values are explained in the legend, and that these fit to those shown in the figure. Please provide statistical testing where applicable. Please avoid the phrase 'independent experiment', but clearly state if these were biological or technical replicates. Please also indicate (e.g. with n.s.) if testing was performed, but the differences are not significant. In case n=2, please show the data as separate datapoints without error bars and statistics. See also:

<http://www.embopress.org/page/journal/14693178/authorguide#statisticalanalysis>

If n<5, please show single datapoints for diagrams. Moreover:

- Please note that the scale bar is missing for figures 2D, 3D, 4F, 5I.
- Please note that scale bar and its definition are missing for figure 5D.
- Thus, please add scale bars of similar style and thickness to microscopic images, using clearly visible black or white bars (depending on the background). Please place these in the lower right corner of the images themselves. Please do not write on or near the bars in the image but define the size in the respective figure legend.
- Please make sure that all the funding information is also entered into the online submission system and that it is complete and similar to the one in the acknowledgement section of the manuscript text file. Presently, Samuel Jordan Graham postdoctoral fellowship and funds from Bloomberg Philanthropies are missing in the submission system. The text in the Comments box needs to be removed, especially since all funders (e.g. Samuel Jordan Graham postdoctoral fellowship) need to be entered separately.
- Please add a paragraph titled 'Biosafety' to the methods section gathering all information on where and how biosafety-relevant experiments with pathogens were performed and that these were approved, and by whom (institution, government).
- Please upload the reagents and tools table as separate file and remove it from the manuscript text. Please also add callouts to the table in the Methods section where appropriate.
- Please add a direct link to the data deposited at BiolImages.
- Please upload the schematic summary figure in jpeg or tiff format (with the exact width of 550 pixels and a height of not more than 400 pixels).

Best,

Referee #1:

The authors have reflected all of my comments finely. They now clearly state that the complementation test by itself does not directly mean that it is lysidine that facilitates the reading of the AUA codons. They also have corrected the bad citation of the paper in which lysidine was first characterized. So, I believe that the manuscript is now suitable for publication as is in the journal.

Referee #2:

The authors have addressed my concerns.

Referee #3:

The points I raised have been suitably addressed.

All editorial and formatting issues were resolved by the authors.

Sean Prigge
Johns Hopkins Bloomberg School of Public Health
Department of Molecular Microbiology and Immunology
615 N. Wolfe St
Baltimore, MD 21205
United States

Dear Dr. Prigge,

I am very pleased to accept your manuscript for publication in the next available issue of EMBO reports. Thank you for your contribution to our journal.

Yours sincerely,
